# Geometry of visuospatial working memory information in miniature gaze patterns

**Juan Linde-Domingo** ⬡ [1,2,3,4] ✉ **& Bernhard Spitzer** ⬡ [1,2] ✉

Stimulus-dependent eye movements have been recognized as a potential confound in decoding visual working memory information from neural signals. Here we combined eye-tracking with representational geometry analyses to uncover the information in miniature gaze patterns while participants ($n = 41$) were cued to maintain visual object orientations. Although participants were discouraged from breaking fixation by means of real-time feedback, small gaze shifts (<1°) robustly encoded the to-be-maintained stimulus orientation, with evidence for encoding two sequentially presented orientations at the same time. The orientation encoding on stimulus presentation was object-specific, but it changed to a more object-independent format during cued maintenance, particularly when attention had been temporarily withdrawn from the memorandum. Finally, categorical reporting biases increased after unattended storage, with indications of biased gaze geometries already emerging during the maintenance periods before behavioural reporting. These findings disclose a wealth of information in gaze patterns during visuospatial working memory and indicate systematic changes in representational format when memory contents have been unattended.

Working memory (WM) enables observers to actively keep stimulus information 'on the mind' for upcoming tasks. A key question in understanding WM function is which aspects of a task-relevant stimulus it retains and in which format(s). On removal of a stimulus from sight, sensory systems briefly retain a detailed sensory memory of the just-removed (for example, visual) input. Without active maintenance, these rich 'photographic' memories decay rapidly in a few hundreds of milliseconds[1] (but see ref. 2). It is widely assumed that only a limited amount of information can be accurately maintained in WM[3–6]. However, despite intense research, the very nature of the information that WM maintains remains poorly understood.

In neuroscientific experiments examining the representation of visual WM information in the brain, often only a single stimulus feature needs to be reported after a delay (for example, the orientation of a visual grating)[7–10]. At one extreme, such tasks could be solved by sustaining a concrete visual memory of the stimulus and its visual details. Various human neuroimaging studies have shown that WM contents can be decoded from early visual cortices[8,10,11], which seems consistent with storage in a sensory format (but see ref. 12). At the other extreme, many WM tasks can also be solved with a high-level abstraction of the task-relevant stimulus parameter only, such as its orientation, speed or colour[12–15]. Such abstractions may also be recoded into pre-existing categories (such as 'left', 'slow' or 'green')[16,17], which may result in memory reports that are biased[18] but still sufficient to achieve one's behavioural goals. Abstraction may render working memories more robust, afford transfer across tasks and massively reduce the amount of information that must be maintained[17,19,20].

However, progress in understanding the temporal dynamics of WM abstraction has thus far been limited. A few studies have examined the extent to which neural WM representations generalize (or not)

[1]Research Group Adaptive Memory and Decision Making, Max Planck Institute for Human Development, Berlin, Germany. [2]Center for Adaptive Rationality, Max Planck Institute for Human Development, Berlin, Germany. [3]Mind, Brain and Behavior Research Center, University of Granada, Granada, Spain. [4]Department of Experimental Psychology, University of Granada, Granada, Spain. ✉e-mail: lindedomingo@mpib-berlin.mpg.de; spitzer@mpib-berlin.mpg.de

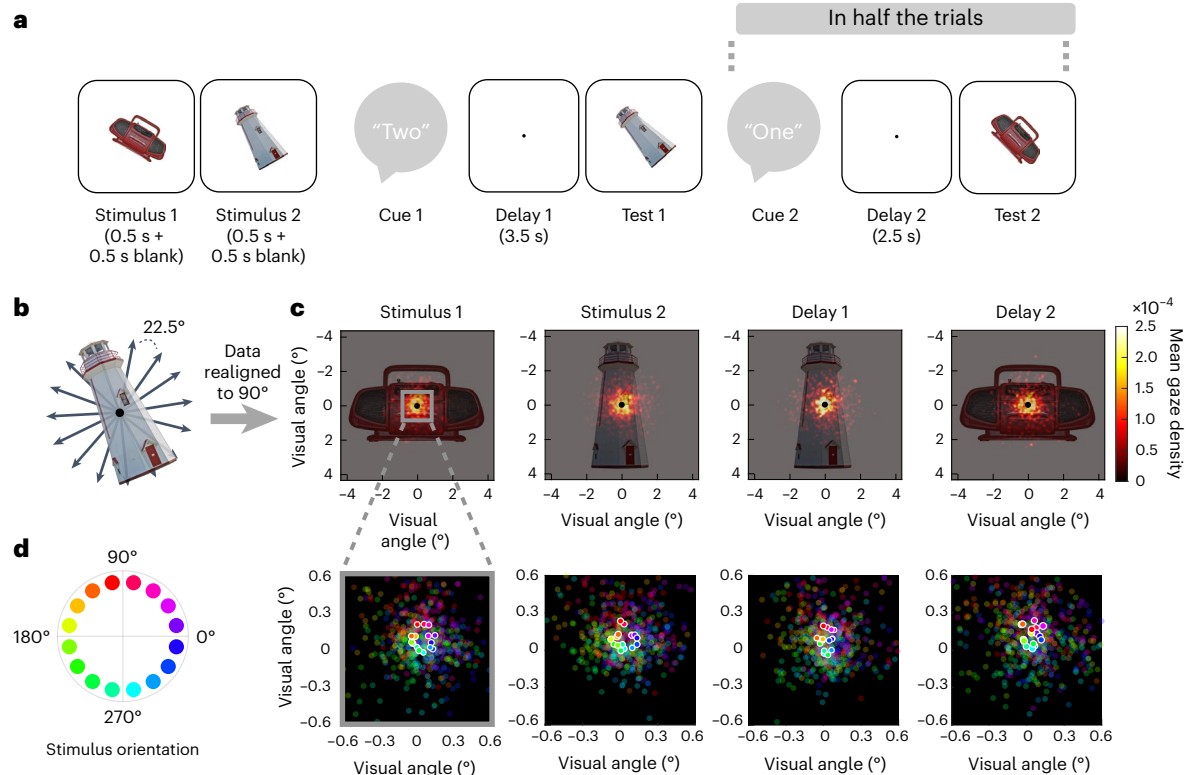

**Fig. 1 | Eye-tracking during WM for visual object orientation. a**, Example trial. After presentation of two randomly oriented objects (Stimulus 1 and 2), an auditory cue indicated which of the two stimulus orientations was to be remembered after an unfilled retention interval (Delay 1). At test, participants were asked to rerotate the probe stimulus to its memorized orientation (2-AFC). In half the trials (randomized), participants were subsequently cued to also remember the orientation of the previously uncued stimulus after another retention period (Delay 2 and Test 2). Participants were instructed to fixate a centred dot throughout the delay periods, and fixation breaks were penalized with closed-loop feedback from online eye-tracking (Methods). **b**, Across trials, the orientations of the memory items were randomly varied around the full circle (in 16 steps, excluding cardinal orientations). **c**, Illustration of the gaze data relative to visual stimulus size. Heatmaps show gaze densities (aggregated across participants) after aligning (rerotating) the data from each trial to the object's upright (90°) position. Panels show the densities aggregated over different trial periods (see **a**), with rotational alignment to the currently relevant stimulus orientation (see corresponding example objects in **a**). **d**, Mean gaze positions (without rotational alignment) during the trial periods in **c** for the 16 different orientations of the currently relevant stimulus (see colour legend at left). Plots show magnification (5×) of the central display area outlined in **c**, left. Saturated dots, mean, unsaturated dots, individual participants. Stimulus images are from ref. 71, the Bank of Standardized Stimuli (BOSS), and licensed under CC-BY-SA 3.0.

across different stimulus inputs[12,14,15,21] and/or become categorically biased[22–24]. In humans, these studies used functional imaging, which lacks the temporal resolution to disclose rapid format changes, or electroencephalography (EEG)/magnetoencephalography, which often can decode the task-relevant stimulus information only during the first 1–2 seconds of unfilled WM delays[22,25–28]. Here, we used a different approach that leverages the finding that subtle ocular activity (for example, microsaccades)[29,30] can reflect attentional orienting during visuospatial WM tasks[31]. Although traditionally considered a confound that experimenters seek to avoid, small gaze shifts can reflect certain types of visuospatial WM information with greater fidelity than EEG/magnetoencephalography recordings[32,33] and even throughout prolonged WM delays, which opens new avenues for tracking dynamic format changes.

On the basis of previous behavioural and theoretical work, we hypothesized that the level of abstraction in WM may change when the to-be-maintained information has been temporarily unattended. While unattended, WM contents cannot easily be decoded with neuroimaging approaches (but see ref. 21), and the neural substrates of unattended storage remain disputed[25,34,35]. Behaviourally, however, temporary inattention renders working memories less precise[36,37] and more categorically biased[38], which may indicate increased abstraction of the WM content[39]. Physiological evidence for when and how such modifications may occur during WM maintenance is still lacking.

Here we recorded eye movements while participants memorized the orientations of rotated objects in a dual retro-cue task (Fig. 1a). With such a task layout, it is commonly assumed that the initially uncued information is unattended (or deprioritized) in WM, and the cued information is in the focus of attention[28,40,41]. Representational geometry analyses borrowed from neuroimaging[42] allowed us to track with high temporal precision whether orientation encoding in gaze patterns was object-specific (indicating a concrete visual memory), object-independent (indicating more generalized/abstract task coordinates) and/or categorically biased throughout the different stages of the task. Participants were encouraged to keep fixation through online feedback (closed loop) to restrict eye movements to small and involuntary gaze shifts.

We found that despite this fixation monitoring, miniature gaze patterns clearly encoded the cued stimulus orientation throughout the WM delays. Although the orientation encoding was object-specific at first (indicating attentional focusing on concrete visual details), its format rapidly became object-independent (generalized/abstract) when another stimulus was encoded or maintained in the focus of attention. We further found that temporary inattention increased repulsive cardinal bias in subsequent memory reports, with some evidence for such biases already emerging during the delay periods in the geometry of gaze patterns. Together, our findings indicate adaptive format changes during WM maintenance within and outside the focus of attention and highlight the utility of detailed gaze analysis for future work.

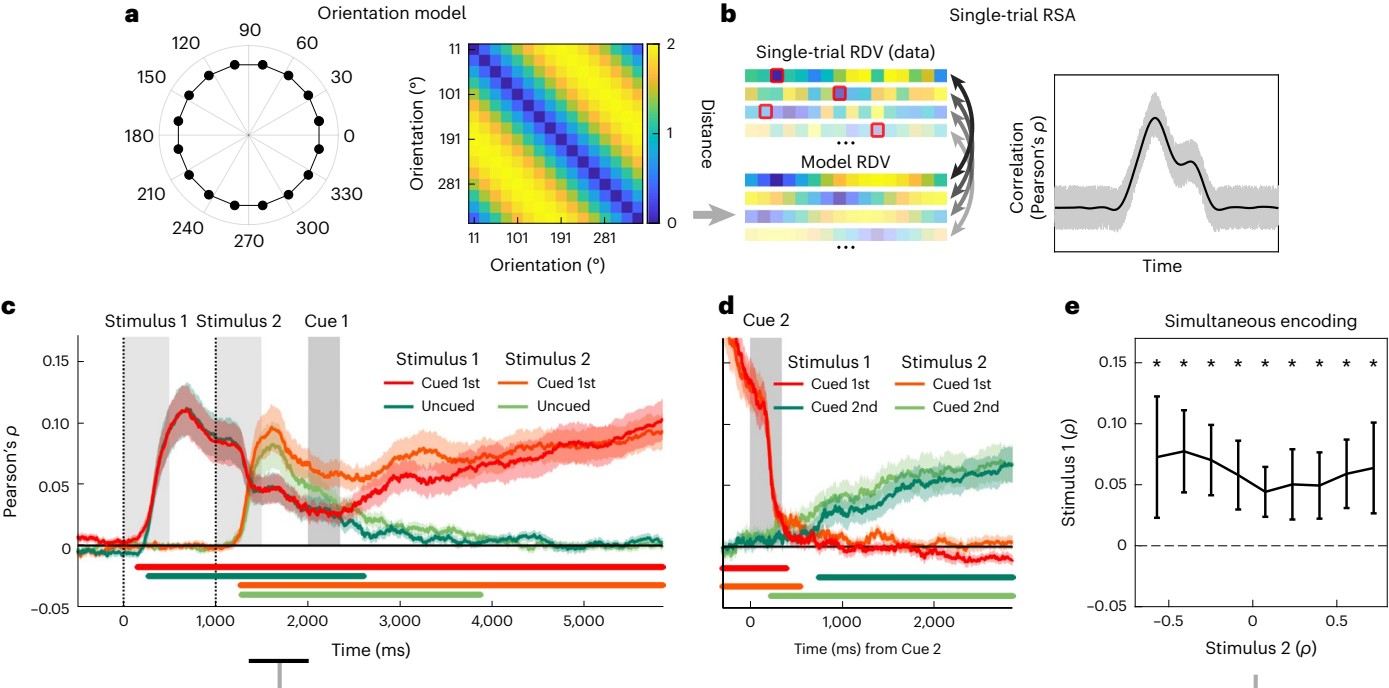

**Fig. 2 | Single-trial RSA and orientation encoding time courses. a**, Left, stimulus orientations (Fig. 1b) plotted as points on a circle. Right, pairwise Euclidean distances (a.u.) between the circle points in **a**. **b**, RSA was performed at the single-trial level. Left, RDV of the Euclidean distances between the gaze position on the current trial (current orientation highlighted by red squares) and the mean gaze positions associated with the 16 different orientations (trial averages with the current trial data held out). Gaze RDVs were obtained at each time point and correlated trial by trial with the corresponding model RDV (that is, the distances predicted by the circular orientation model in **a**), yielding a time course of orientation encoding for each trial (Methods). Right, the single-trial approach yields a mean time course of orientation encoding as would be obtained with conventional RSA (black lines) while also retaining trial-by-trial variability (grey shadings). **c**, Mean encoding of the two stimulus orientations, shown separately for when Stimulus 1 or Stimulus 2 was cued for Test 1 (Fig. 1a). Vertical dotted lines indicate the times of stimulus onset. Grey vertical bars indicate duration of stimuli and auditory cues ('one'/'two'). Coloured shadings show standard error of the mean (s.e.m.). Coloured marker lines at the bottom indicate significant orientation encoding (display threshold $P_{cluster} < 0.0125$ to account for testing in four conditions). **d**, Same as **c**, for the second delay period (Delay 2; Fig. 1a, right). Note that the strong precue effect is explained by residual eye movements related to Test 1. **e**, Single-trial analysis of Stimulus 1 orientation encoding concurrent to encoding the orientation of Stimulus 2 (see time window outlined by black marker in **c**, bottom). Trials were binned according to orientation encoding strength for Stimulus 2 (x axis; mean values of bins averaged over participants), with orientation encoding strength for Stimulus 1 plotted on the y axis. Data are presented as mean values ± s.e.m. (n = 41 participants). Asterisks indicate significant differences from 0 (P < 0.05, Bonferroni-corrected for the number of bins). Significant encoding of the orientation of Stimulus 1 was evident at each bin (all $t(40) > 3.116$, all $P < 0.003$ (uncorrected), all $d > 0.487$, t-tests against zero, two-tailed), indicating that gaze position carried information about both stimulus orientations simultaneously. Stimulus images adapted from ref. 71 under a Creative Commons licence CC BY 4.0.

## Results

Participants (n = 41) performed a cued visual WM task (Fig. 1a) while their gaze position was tracked. On each trial, two randomly oriented stimuli (pictures of real-world objects) were sequentially presented (each for 0.5 s followed by a 0.5 s blank screen), after which an auditory 'retro'-cue (Cue 1) indicated which of the two orientations was to be remembered after a delay period (Delay 1, 3.5 s) at Test 1. On half the trials (randomly varied), Test 1 was followed by another retro-cue (Cue 2) and another delay period (Delay 2, 2.5 s), after which participants were required to also remember the orientation of the other, previously uncued stimulus (Test 2).

### Behavioural accuracy

At each of the two memory tests, the probed stimulus was shown with a slightly altered orientation (±6.43°), and participants were asked to rerotate it to its previous orientation by means of button press (two-alternative forced choice, 2-AFC). As expected, the percentage of correct responses was descriptively higher on Test 1 (mean = 73.41%, standard deviation (s.d.) = 6.42%) than on Test 2 (mean = 66.62%, s.d. = 5.78%). Further, the second presented orientation (Stimulus 2) was remembered better (mean = 70.99%, s.d. = 7.07%) than the first

presented orientation (Stimulus 1; mean = 69.04%, s.d. = 6.79%). A 2 × 2 repeated-measures analysis of variance (ANOVA) with the factors Test (1/2) and Stimulus (1/2) confirmed that both these effects were significant ($F(1,40) = 95.396$, $P < 0.001$, eta squared ($\eta^2$) = 0.521 and $F(1,40) = 13.319$, $P < 0.001$, $\eta^2 = 0.043$)), whereas there was no significant interaction between the two factors ($F(1,40) = 3.681$, $P = 0.062$, $\eta^2 = 0.008$).

The effects of presentation and testing order may both be attributed to task periods since stimulus presentation during which the other stimulus was to be attended, either for perceptual processing (Stimulus 2) or for cued maintenance and reporting (Delay 1 and Test 1). We may combine these two factors into the 'mnemonic distance' of a stimulus, which in our experiment had four levels (from shortest to longest: Stimulus 2 at Test 1, Stimulus 1 at Test 1, Stimulus 2 at Test 2 and Stimulus 1 at Test 2). The behavioural accuracy results were compactly described as a monotonic decrease across these distance levels ($t(40) = -9.404$, $P < 0.001$, Cohen's d = -1.469, 95% confidence interval (CI) (−0.038, −0.024); t-test of linear slope against zero), as would be expected if processing Stimulus 2 temporarily withdrew attention from the memory of Stimulus 1, similar (and additive) to the withdrawal of attention from the uncued item during Delay 1 and Test 1.

## Object orientation was reflected in miniature gaze patterns

We informed participants that their gaze would be monitored to ensure that they constantly fixated a centrally presented dot throughout the task. To enforce this, we provided real-time feedback (closed-loop) when fixation was lost (Methods). Figure 1c shows the participants' gaze distribution in relation to stimulus size after rotating the trial data to the respective object's real-world (upright) orientation (for similar approaches, see refs. 43,44). Despite this rotation alignment, the gaze density was concentrated narrowly (mostly in a <1° visual angle) at centre during both the stimulus and the delay periods (Fig. 1c). The instructions and online feedback thus proved effective in preventing participants from overtly gazing at the location of the objects' peripheral features (such as, for example, the spire of the lighthouse in Fig. 1a). However, inspecting the participants' average gaze positions for each stimulus orientation (without rotational alignment) disclosed miniature circle-like patterns (Fig. 1d), indicating that miniscule gaze shifts near fixation did carry information about the objects' orientation (for related findings with other stimulus materials, see refs. 32,33).

For quantitative analysis of the orientation encoding in gaze, we used an approach on the basis of representational similarity analysis (RSA)[42]. Specifically, we examined the extent to which the gaze patterns showed the characteristic Euclidean distance structure of evenly spaced points on a circle (Fig. 2a). We implemented RSA on the single-trial level (Fig. 2b) by correlating for each trial the model-predicted distances with the vector of gaze distances between the current trial and the trial average for each stimulus orientation. The procedure yields a cross-validated estimate of orientation encoding at each time point for every trial (Methods).

We first inspected the mean time courses of orientation encoding (averaged over trials) during stimulus presentation. We observed robust encoding of stimulus orientation from about 500 ms after stimulus onset for both Stimulus 1 and Stimulus 2 (both $P_{cluster}$ < 0.001, cluster-based permutation tests; Methods). The encoding of either stimulus orientation peaked at ~650 ms (that is, only after the stimuli's offset; Fig. 2c), after which it slowly decayed.

### Concurrent encoding of Stimulus 1 and Stimulus 2

Although gaze data are only two-dimensional, we found that while encoding the second presented orientation (Stimulus 2), the gaze pattern also continued to carry information about the first-presented orientation (Stimulus 1; see Fig. 2c). Such a concurrency in the average time courses may have arisen if one of the orientations was encoded on some trials and the other orientation on others. Alternatively, however, the pattern may indicate that gaze encoded both orientations simultaneously (that is, additively, on the same trials). To shed light on this, we capitalized on our single-trial approach (Methods and Fig. 2b) and binned each participant's trials according to how strongly the orientation of Stimulus 2 was encoded between 250 and 1,000 ms after Stimulus 2 onset (Fig. 2e). If encoding of the two orientations had alternated between different trials, we would expect a negative relationship with the encoding of the orientation of Stimulus 1 in the same time window. However, we found no significant relationship ($t(40) = -1.368$, $P = 0.18$, $d = -0.214$, 95% CI (−0.006, 0.001); linear trend analysis). What is more, the encoding of Stimulus 1's orientation was significantly above chance even on those trials on which the encoding of Stimulus 2's orientation was maximally strong ($t(40) = 3.656$, $P = 0.01$, $d = 0.571$, 95% CI (0.029, 0.099); $t$-test against 0). Together, these results indicate that small shifts in 2D gaze space carried information about the two stimulus orientations simultaneously, on the same trials.

### Encoding of the cued orientations throughout the delay periods

Our main interest was in how gaze patterns reflected information storage during the unfilled delay periods (Delay 1 and 2; Fig. 1a). During Delay 1, about 500 ms after auditory cueing (Cue 1), the encoding of the cued orientation ramped up and continuously increased in strength until the time of Test 1, whereas the encoding of the uncued orientation slowly returned to baseline (Fig. 2c). During Delay 2 (which occurred in half the trials), a similar ramping-up pattern was observed for the second-cued orientation (which was previously uncued; Fig. 2d; both $P_{cluster}$ < 0.001). Thus, miniature gaze deflections robustly encoded the currently cued (or 'attended') memory information during the two delay periods in a ramp-up fashion that resembled the encoding of WM information in neural recordings (for example, in monkey prefrontal cortex)[45,46].

### Object-specific versus object-independent orientation encoding

We next examined more closely the format(s) in which the gaze patterns reflected the WM information. A priori, memory reports in our task could be on the basis of a concrete visual memory of the presented stimulus, but they could also be on the basis of a mental abstraction of orientation: for example, in terms of directional spatial coordinates. To the extent that the small eye movements during WM maintenance reflected mental focusing on concrete visual features (for example, the location of a specific point on the object's contour), we expect the orientation encoding in gaze to be object-specific: that is, not fully transferable between different objects. In contrast, an abstraction of orientation (for example, in terms of a direction in which any object may point with its real-world top) should be reflected in gaze patterns that are object-independent and transferable.

We examined object specificity by comparing the orientation encoding in gaze distances within objects (Fig. 3a, left) with that in gaze distances between objects (Fig. 3a, right). On stimulus presentation, the orientation encoding in gaze patterns was object-specific, in that within-objects encoding clearly exceeded between-objects encoding (Fig. 3c; all $P_{cluster}$ < 0.012). For Stimulus 1, the object specificity diminished abruptly after ~1,300 ms (when the gaze patterns began to also

**Fig. 3 | Object-specific versus object-independent orientation encoding. a**, Orientation model analogous to Fig. 2a but extended to separately examine orientation encoding within (left) and between (right) objects. Unsaturated colours delineate distances that are excluded from the respective submodel. The degree of object specificity is inferred from the extent to which within-objects encoding is stronger than between-objects encoding. **b**, Difference in orientation encoding within objects compared to between objects for the cued stimulus during each delay period. Data are averaged from cue onset to the end of the delay. The four conditions are sorted by the time distance from stimulus presentation ('mnemonic distance' from lowest to highest). Grey dots show individual participant results, and trend lines show linear fits. Box plots show group means ± s.e.m. (boxes) and ± s.d. (whiskers), $n = 41$ participants. **c**, Orientation encoding time courses within and between objects during Delay 1, shown separately for when the stimulus was cued (pink and purple) or uncued (light and dark green). Top, Stimulus 1. Bottom, Stimulus 2. Coloured shadings show s.e.m. Coloured marker lines at the bottom indicate significant object specificity in terms of stronger orientation encoding within than between objects (display threshold $P_{cluster}$ < 0.0125). Otherwise, same conventions as Fig. 2c. **d**, Same as **c**, but for the second delay period (Delay 2). **e,f**, Bayes Factor ($BF_{01}$, one-tailed) analysis of the difference between within- and between-objects encoding. BF time courses are shown for the cued orientation in the respective task periods (**e**, Delay 1; **f**, Delay 2). Negative values on the log scale ($y$ axis) indicate stronger evidence for object-specific encoding (within > between) than for object-independent encoding (within ≤ between); positive values indicate the opposite. Results are shown for periods of significant overall orientation encoding (see Fig. 2) where the comparison of within- and between-objects encoding is meaningful. The data were smoothed with a 50 ms Gaussian kernel before this analysis. Saturated colours indicate stronger-than-anecdotal evidence (logBF < −1.1 or > 1.1, which corresponds to BF < 1/3 or > 3).

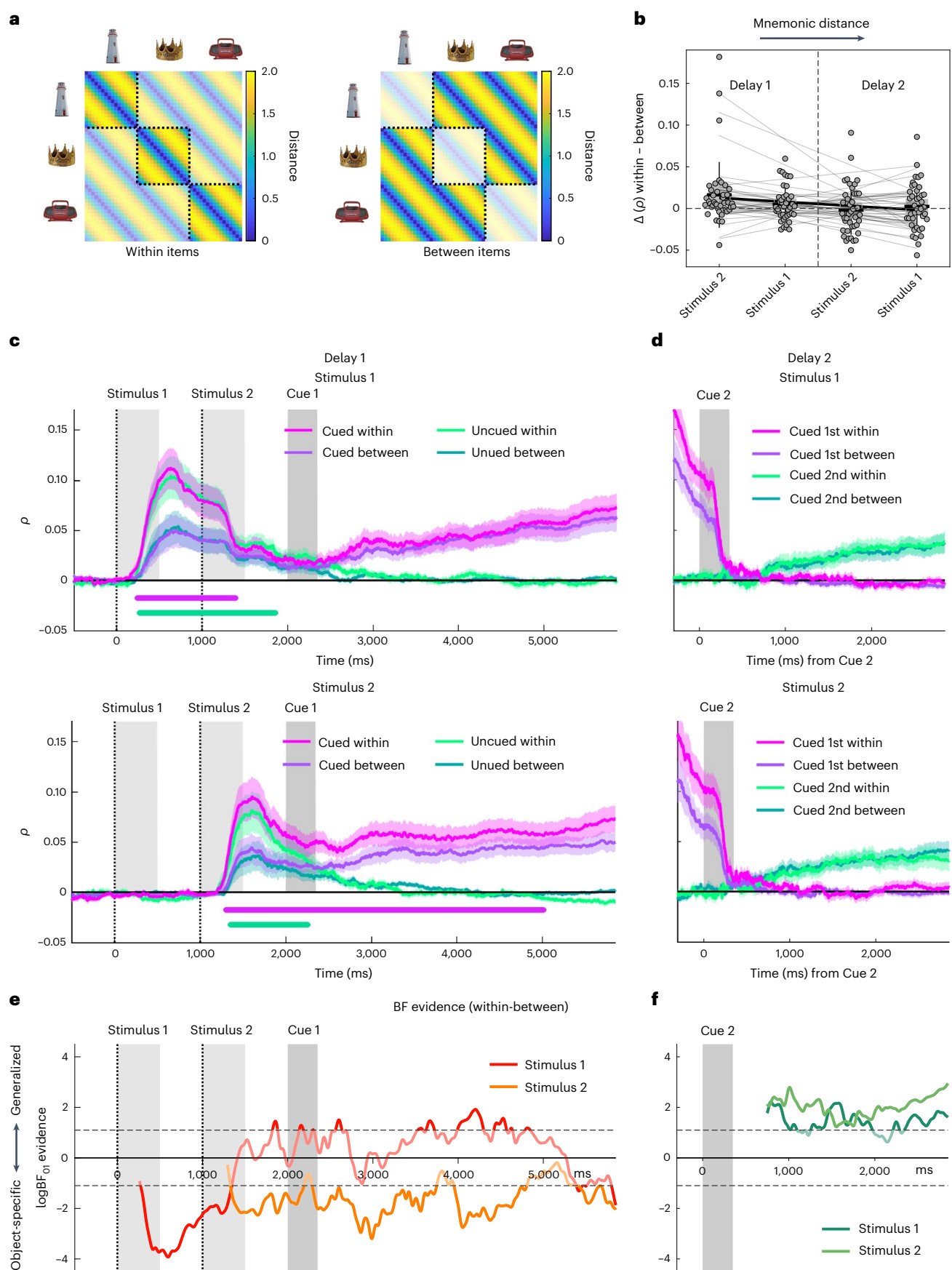

encode Stimulus 2) and changed to a more object-independent format for the remainder of the trial epoch (Fig. 3c, top). For Stimulus 2, in contrast, when cued for Test 1, the object specificity decayed less and was sustained throughout most of Delay 1 (Fig. 3c, bottom). Later, in Delay 2, no object specificity was evident for either stimulus (no $P_{cluster} < 0.60$). Figure 3e,f summarizes the temporal evolution of object independence in terms of Bayes factors, showing the swift change of Stimulus 1 encoding from object-specific ($BF_{01} < 1/3$) towards object-independent ($BF_{01} > 3$) at the time of Stimulus 2 encoding, whereas the encoding of Stimulus 2 retained object specificity during Delay 1 (Fig. 3e). In Delay 2, after unattended storage throughout Delay 1, the orientation encoding in gaze had become object-independent for both stimuli (Fig. 3f).

Focusing on the delay periods, we examined whether the object specificity of cued orientation encoding differed between the two delay periods (Delay 1 or 2) and/or between the first and second presented stimulus (Stimulus 1 or 2). A 2 × 2 repeated-measures ANOVA on the difference in encoding strength (within- minus between-objects, averaged across the respective delay periods) showed a main effect of delay period (Delay 1/2; $F(1,40) = 6.204$, $P = 0.017$, $\eta^2 = 0.064$) indicating greater object specificity during Delay 1 but no effect of presentation order (Stimulus 1/2; $F(1,40) = 0.985$, $P = 0.327$). There was also moderate interaction between the two factors ($F(1,40) = 4.466$, $P = 0.041$, $\eta^2 = 0.027$), reflecting that the difference between the two delays was stronger for Stimulus 2. Again, we also inspected these results in terms of the mnemonic distance from stimulus presentation, that is, the time the orientation in question had been unattended while focusing on the other orientation (Fig. 3b). Indeed, this analysis confirmed a decrease in object specificity with increasing mnemonic distance ($t(40) = -2.473$, $P = 0.018$, $d = -0.386$, 95% CI ($-0.010$, $-0.001$); $t$-test of linear slope against zero).

Together, these results showed that unlike during perceptual processing, gaze patterns during the delay periods reflected WM information in more generalized (or abstract) coordinates and that the level of this abstraction increased after periods of temporary (or partial) inattention.

### Cardinal repulsion bias in gaze patterns and behaviour

In studies of WM for stimulus orientation (for example, of Gabor gratings) it is commonly observed that behavioural reports are biased away from the cardinal (vertical and horizontal) axes[18,22]. We asked (1) whether such a repulsive cardinal bias also occurred with our rotated object stimuli, (2) whether the strength of bias was modulated by periods of inattention[38] and (3) the extent to which such bias was already expressed in the geometry of the miniature gaze patterns observed during the delay periods.

To model bias in behaviour, we used a geometrical approach that quantifies bias as a mixture of a perfect (unbiased) circle (Fig. 4a, middle) with perfect (fully biased) square geometries (Fig. 4a, leftmost and rightmost; see Methods, 'Behavioural modelling' for details). Intuitively, the mixture parameter $B$ quantifies the extent to which the reported orientations were repulsed away from the cardinal axes,

with $B > 0$ indicating repulsion (that is, cardinal bias), $B = 0$ no bias and $B < 0$ attraction.

Fitting the model to participants' behavioural responses (Fig. 4b, left and middle), we observed values of $B > 0$ (grand mean, 0.124, s.d. = 0.092) in both memory tests (Test 1 and 2) and for both orientations (Stimulus 1 and 2; all $B > 0.071$; all $t(40) > 4$, all $P < 0.001$; $t$-tests against 0). Thus, participants overall showed a repulsive cardinal bias, which replicates and extends previous work with simpler stimuli (such as gratings)[18,22]. A 2 × 2 repeated-measures ANOVA showed a main effect of test (Test 1/2; $F(1,40) = 19.743$, $P < 0.001$, $\eta^2 = 0.144$) indicating a stronger bias on Test 2 and a main effect of presentation order (Stimulus 1/2; $F(1,40) = 4.669$, $P = 0.037$, $\eta^2 = 0.024$) with no interaction between the two factors ($F(1,40) = 1.083$, $P = 0.304$). The overall pattern could again be described compactly as an increase in cardinal bias with increasing mnemonic distance from stimulus presentation ($t(40) = 5.315$, $P < 0.001$, $d = 0.830$, 95% CI (0.019, 0.043); $t$-test of linear slope against zero; Fig. 4b, left). Thus, we found robust cardinal repulsion in participants' overt memory reports, and this bias increased with periods of unattended storage.

Finally, we addressed the extent to which the cardinal bias was also reflected in the gaze patterns recorded throughout the two delay periods. To do so, our geometric model yields distinctive distance structures for extreme cardinal repulsion ($B = 1$; Fig. 4a, rightmost) and attraction ($B = -1$; Fig. 4a, leftmost), respectively. If the gaze patterns were unbiased, we would expect both these 'square' models to correlate less well with the data than the unbiased ('circle') model with $B = 0$ (Fig. 4a, middle). However, to the extent that the gaze patterns were repulsively biased, we would expect the repulsion model to outperform the attraction model, nearing (or, in the case of extreme bias, even exceeding) the circle model (dashed black in Fig. 4c). Contrasting repulsion and attraction models thus allowed us to quantify the extent of repulsive or attractive bias in the gaze patterns during the delay periods.

Descriptively, the three different models (repulsion, unbiased, attraction) showed only small differences in correlation with the data (Fig. 4c), indicating that the statistical power to detect bias in the gaze data was relatively low (see Methods, 'Model geometries'). Nevertheless, contrasting the repulsion model with the attraction model showed two small clusters ($P_{cluster} = 0.02$ and $P_{cluster} = 0.035$), indicating a repulsive bias, near the end of the delay periods for Stimulus 1 (Fig. 4c, top). A similar tendency for Stimulus 2 failed to reach significance in Delay 2 (Fig. 4c, lower-right; $P_{cluster} = 0.085$, below display threshold) and was absent in Delay 1 (Fig. 4c, lower-left; no cluster-forming time points). A 2 × 2 repeated-measures ANOVA on the difference between repulsion and attraction models (averaged across the last second of the delay periods) showed a main effect of presentation order (Stimulus 1/2; $F(1,40) = 4.561$, $P = 0.039$ $\eta^2 = 0.026$; main effect of Delay 1/2: $F(1,40) = 1.650$, $P = 0.206$; interaction: $F(1,40) < 1$) indicating a stronger repulsive bias for the first presented orientation (Stimulus 1). Complementary analysis in terms of mnemonic distance (Fig. 4b, right) showed a positive trend similar to that for behaviour, albeit only

**Fig. 4 | Cardinal repulsion bias in gaze patterns and behaviour. a**, Model-predicted geometries (top) and RDMs (bottom) associated with different levels of cardinal bias. The mixture parameter $B$ (black arrow axis at the bottom) denotes the level of repulsion ($B > 1$, 'cardinal bias') or attraction ($B < 1$) from/ to cardinal axes, relative to the unbiased circle model ($B = 0$). **b**, Left and middle, results from the model fitted to the behavioural memory reports. Left, estimates of cardinal repulsion bias ($B$) for each stimulus (1/2) and test (1/2), sorted by the distance between stimulus presentation and test. Grey dots show individual participants results and trend lines show linear fits. Box plots show group means ± s.e.m. (boxes) and ± s.d. (whiskers), $n = 41$ participants. Middle, polar plot shows the mean proportions of clockwise reports (green) and the predictions of the fitted model (magenta, $B = 0.12$) for each stimulus orientation. Results are averaged over both stimuli and tests. Dashed black line shows

proportions (50%) expected under an unbiased circular model ($B = 0$) for visual reference. Right, quantification of bias in the gaze patterns associated with the cued stimulus orientation. Shown are the differences in correlation of the gaze patterns with the repulsion model ($B = 1$) compared to the attraction model ($B = -1$), where a positive difference indicates repulsive cardinal bias. Results are shown for the last second of the respective delay period (see **c**). Same plotting conventions as left. **c**, Time course of correlations with the repulsion (red) and attraction (blue) models during the delay periods (same layout as Fig. 3c,d). Coloured shadings show s.e.m. Dashed black line shows correlation with the unbiased circular model ($B = 0$) for visual reference. Red marker lines at the bottom indicate stronger correlation with the repulsion than the attraction model (display threshold $P_{cluster} < 0.05$).

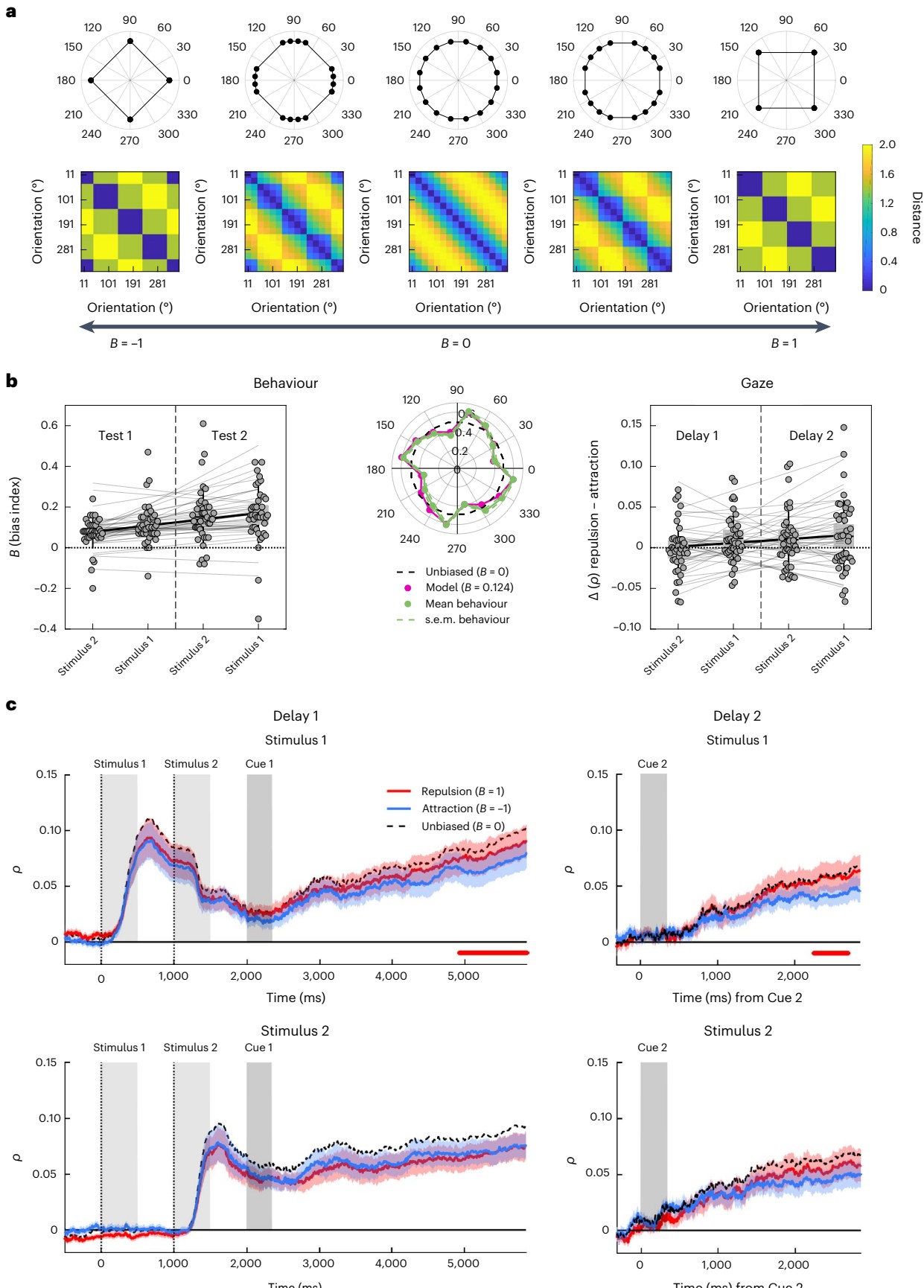

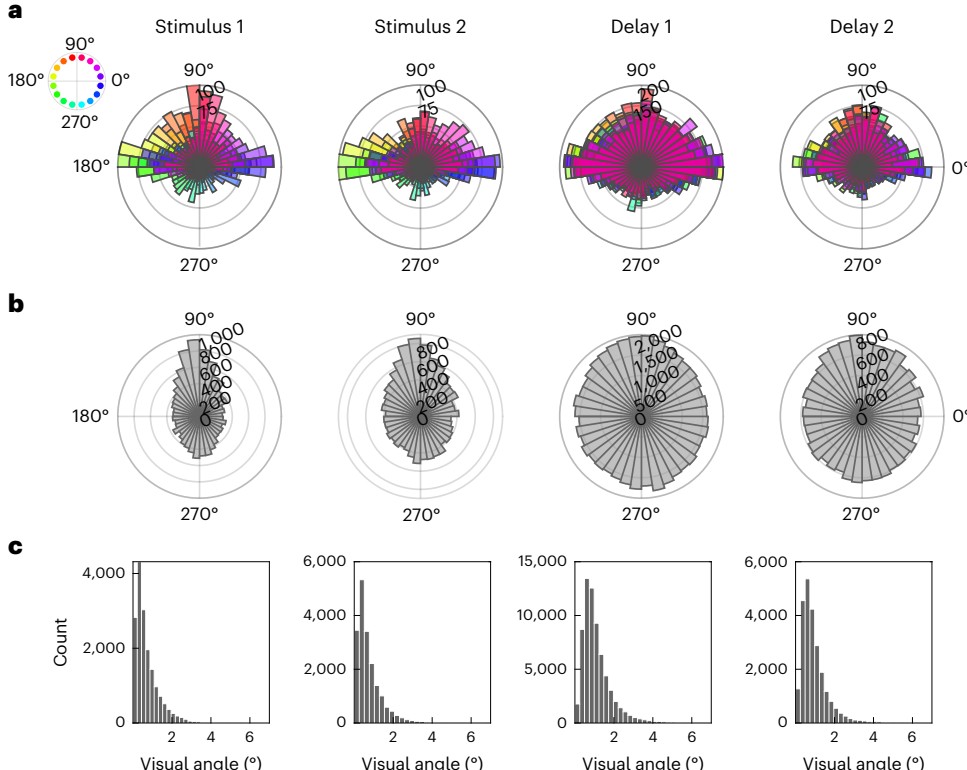

**Fig. 5 | Complementary analysis of microsaccadic activity.** We performed microsaccade detection (Methods) in four time windows, during which orientation encoding was evident in the gaze-position data (Fig. 2c,d): from 300 to 1,000 ms after the onset of each stimulus (Stimulus 1, Stimulus 2), from after Cue 1 until the end of the first delay (Delay 1) and from 300 ms after Cue 2 until the end of the second delay (Delay 2). **a**, Distribution of microsaccade directions (onset to endpoint; Methods) as a function of stimulus orientation (see the colour legend on the left), collapsed across all trials from all participants.

The distributions during the delay periods (Delay 1 and 2) are colour-coded according to the currently cued orientation, respectively. **b**, Distribution of microsaccade directions after alignment (analogous to Fig. 1b,c, top) relative to the objects' upright (90°) position. **c**, Histograms of the sizes (amplitudes) of the saccades detected for analysis in **a** and **b**. Most saccades in each time window were considerably smaller than 2°, in line with an interpretation in terms of microsaccadic activity[47].

at the significance level of a one-tailed test ($t(40) = 1.772$, $P = 0.042$, $d = 0.278$, 95% CI lower bound −0.001; $t$-test of linear slope against zero; one-tailed, hypothesis derived from behavioural result). Together, although the differentiation of models (repulsive, unbiased, attractive) in the gaze data was not as clear-cut as in behaviour (cf. Fig. 4b, right and left), we found indications that the gaze patterns may have carried a repulsive cardinal bias, most evidently during the later portions of the WM delays and after temporary and/or partial inattention to the WM information.

### Orientation-dependent microsaccades

Although our RSA-based approach was designed to characterize the time-varying geometries of aggregate gaze-position patterns (Figs. 2–4), we performed further analysis on request by reviewers to explore whether the findings may indeed be related to microsaccades: that is, small, 'jerk-like'[47] eye movements. Figure 5 illustrates the directions of microsaccades detected after stimulus presentation (Stimulus 1, Stimulus 2) and during the two delay periods (Delay 1, Delay 2), respectively, for each of the 16 orientations of the currently relevant object. The saccade directions in the poststimulus periods correlated positively with stimulus orientation (circular correlation coefficients ($R$): Stimulus 1, $R = 0.089$; Stimulus 2, $R = 0.077$; both, $P < 0.001$). Weakly positive correlations were also evident during the delay periods (Delay 1, $R = 0.01$; Delay 2, $R = 0.03$; both, $P < 0.001$). For further inspection, we again rotated the trial data (analogous to Fig. 1b,c) to illustrate the saccade directions relative to the objects' real-world (upright) orientation. As expected if microsaccades reflected stimulus orientation, in all time

windows, the aligned distributions were not uniform (Rayleigh tests for uniformity: all $z > 34.36$, all $P < 0.001$) but appeared egg-shaped, with a main peak near the object's real-world top (at 90°) and another, smaller peak near the opposite angle (270°, which may reflect 'return' microsaccades to fixation). Together, these complementary results support the idea that the effects observed in our main analyses may have been related to microsaccadic activity during attempted fixation[29,48].

## Discussion

The processing of WM information during delay periods has been studied extensively using neural recordings (for reviews, see refs. 35,49–53). Here, using novel stimulus materials and tailored geometry analyses, we showed that miniature gaze deflections can disclose an array of WM-associated phenomena that to the best of our knowledge was previously only observed in neural signals, including (1) a sustained encoding of the task-relevant stimulus feature, which (2) shows a different format than during perception, can (3) persist while also encoding new perceptual information, (4) ramps up throughout delay periods when relevant for an upcoming test and (5) returns to baseline when uncued (or 'unattended'). Beyond this, the gaze geometries indicated that temporary inattention rendered the WM information more generalized (object-independent) and potentially more categorically biased. These format changes during maintenance were similarly observed when attention to the memorandum was withdrawn by explicit retro-cueing or by presenting further WM information.

Behaviourally, our results replicate and extend previous findings that temporary inattention renders working memories less precise

and more biased[37,38]. The eye-tracking results shed light on the temporal unfolding of potentially underlying format changes during WM storage. The gaze patterns during perceptual processing were clearly object-specific, indicating a focus on concrete visual details. When the last-seen stimulus (Stimulus 2) was immediately cued (with an auditory retro-cue that offered no visual distraction), some of this object specificity was sustained throughout the ensuing WM delay. In contrast, for the first-presented stimulus (Stimulus 1), the object specificity dropped abruptly as soon as Stimulus 2 processing commenced. Of note, Stimulus 1 encoding did continue throughout Stimulus 2 processing. However, its format changed to object-independent (or 'abstract') during the object-specific (or 'concrete') encoding of Stimulus 2—as if the memory of Stimulus 1 was reformatted to 'evade' the format of the currently perceived stimulus. Subsequent cueing did not revert this effect, nor did we find any re-emergence of object specificity after unattended storage, for either stimulus, in the second delay. Together, these results support the idea that temporary (or partial) inattention may render the task-relevant WM information (here, orientation) increasingly less 'concrete' (visual-sensory) and more generalized or 'abstract'.

Further support for this idea comes from our analysis of the cardinal repulsion bias. In parallel with the object independence of gaze patterns, the repulsive cardinal reporting bias increased with the time a given stimulus had been temporarily (or partially) unattended. Repulsive-orientation bias in WM tasks has been explained, for example, by efficient coding principles, in terms of relatively finer tuning to cardinal orientations, reflecting their relative prevalence in natural environments[18]. An alternative framing of the cardinal repulsion biases in our experiment with real-life objects could be in terms of more explicit semantic categorization (for example, 'left'/'right' and 'up'/'upside-down')[17,54]. The results may thus also reflect increased reliance on semantics[39] and/or (pre)verbal labels when restoring information from unattended storage[34], which would be in line with a higher level of abstraction. Although our geometrical analysis approach is agnostic to the mechanistic cause of cardinal biases, we found some indications that they were also evident in biased gaze geometries during the stimulus-free retention periods (for related findings in neuroimaging, see refs. [22,23,55]). The latter result was statistically weak and should be revisited in future work, possibly under conditions that induce even stronger biases in behaviour (for example, higher WM loads)[18].

A remarkable aspect of our results is the small amplitude of the eye movements that disclosed such rich information. The mass of the raw position samples in our analysis were within a <1° visual angle around fixation (Fig. 1c). A discernible 'circular' structure in averaged data points (Fig. 1d) measured only ~0.2–0.3° in diameter, which is near the eye-tracker's accuracy limit, and was only a fraction of the memory items' physical size. Together with our online fixation control (Methods), these descriptives render it unlikely that our results were attributable to reflexive saccades to the location of peripheral stimulus features. Further analysis (Fig. 5) indicated that the findings more likely reflect microsaccadic activity during attempted fixation. Systematic microsaccade patterns have previously been linked to covert spatial attention[29,48], indicating that in the present context, they might have reflected mental orienting towards a spatial coordinate or direction[30,31]. Together, our results indicate that participants generally oriented attention towards the objects' real-life 'top', but with varying degrees of bias towards specific object features (resulting in object-specific orientation patterns) and/or away from cardinal axes (resulting in cardinal repulsion).

Under a view of the miniature gaze patterns reflecting covert spatial attention, our analysis tracked with high temporal resolution the time course of attention allocation to WM information in a dual retro-cue task. Before cueing, encoding a new stimulus (Stimulus 2) did not immediately eradicate or replace the attentional orienting

to the previous stimulus (but did change its qualitative format; see above). At face value, the temporary simultaneity of both WM contents (Fig. 2e), in a putative index of attention, might seem to be at odds with the idea of an exclusive single-item focus of attention in WM[56,57] (but see refs. [58,59]). However, another possible interpretation is that the (re-)allocation of attention to different stimuli (or tasks) in WM may take time to complete. For instance, the encoding of the uncued stimulus fully returned to baseline only ~0.5–1.5 s after the cue, which is broadly consistent with previous behavioural and EEG work on the time course of WM-cueing effects[27,60–62]. Compared to this, the reformatting into a more generic, object-independent format was rapid, both for Stimulus 1 when encoding Stimulus 2 (see above) and for Stimulus 2 itself when it was uncued. Consistent with these results, a recent study found that low-level perceptual bias induced by concurrent WM information (cf. ref. [63]) dissipated quickly with new visual input[44]. These findings are in line with adaptive format changes in WM, potentially providing fast protection from interference beyond the overall reallocation of attention between different stimuli and/or tasks.

Previous WM studies using retro-cues yielded mixed results about potential costs for the first of two successively presented stimuli. Using visual retro-cues, one study found no differences between visual gratings presented first or second in either behaviour or functional imaging-decoding during the WM delay[8], which has been taken as evidence that intervening stimuli may cause little to no interference for visual WM representations[9]. Another study, using visual retro-cues with tactile WM stimuli, did find lower performance for the first stimulus[27], a finding we replicated here with auditory cueing of visual WM information. One possibility is that different-modality cues (for example, auditory when the WM stimuli were visual) interfere less with the short-term memory of the last-presented stimulus than same-modality cues would (for example, visual cues with visual WM stimuli)[38]. Different-modality cues may thus leave the memory trace of the last-presented stimulus more intact compared to the first stimulus (which is always followed by the same-modality input of the second stimulus). This aside, the format changes induced by the intervening stimulus were qualitatively similar to those after unattended storage, in line with a common explanation in terms of temporarily withdrawn attention.

Our findings of increasingly more object-independent gaze geometries do not rule out that the brain may maintain detailed visual memories in ways that would not register in eye tracking. More generally, we can only speculate whether the minuscule eye movements observed in our experiment played a functional role or whether they were merely epiphenomena of other processes. We consider it possible that our paradigm promoted aspects of WM-related processing to become visible at the surface of ocular activity, but that the ocular activity itself may have had little or no direct role in the WM processing proper (for related discussion, see refs. [30,47,64]; but see refs. [65,66] for a role of eye movements in episodic memory retrieval). This speculation also takes note of several recent failures to decode visuospatial WM information from eye tracking, most notably in control analyses supplemental to neural decoding, where systematic eye movements were ruled out as a potential confound[12,67–69] (but see refs. [32,33,70]). At the same time, our findings sound a cautionary note that stimulus-dependent eye movements in visual WM tasks can be very small, hard to prevent, persistent and, above all, informative.

In summary, despite discouraging participants from eye motion through closed-loop fixation control, we found the orientation of visual objects robustly reflected in miniature gaze patterns during cued WM maintenance. The geometry of the gaze patterns underwent systematic changes, indicating that temporary inattention increased the level of abstraction (and categorical bias) of the information in WM. Stimulus-dependent eye movements may not only pose a potential confound but also be a valuable source of information in studying visuospatial WM.

## Methods

### Participants

Fifty-five participants (31 female, 24 male, mean age 26.95 ± 3.98 years) took part in the experiment. Forty-four of the participants were recruited from a pool of external participants, and 11 were recruited internally in the Max Planck Institute for Human Development. All participants were blind to our research questions, and all of them received compensation of €10 per hour plus a bonus on the basis of task performance (€5 bonus if four out of five randomly selected memory reports were correct). Written informed consent was obtained from all participants, and all experiments were approved by the ethics committee of the Max Planck Institute for Human Development. Two participants (both wearing glasses) were excluded because of difficulties in acquiring a stable eye-tracking signal, and one participant was excluded because she reported feeling unwell during the experimental session. Of the remaining participants, we excluded $n = 9$ for failing to perform above chance level in each of the two memory tests ($P < 0.05$, Binomial test against 50% correct responses). Finally, after preprocessing the eye-tracking data, we excluded $n = 2$ participants for whom more than 15% of the data had to be rejected because of blinks and other recording artefacts. After this, $n = 41$ participants remained for analysis.

### Stimuli, task and procedure

Nine colour photographs of everyday objects from the BOSS database[71] (candelabra, table, outdoor chair, crown, radio, lighthouse, lamppost, nightstand, gazebo) were used as stimuli. All objects were cropped (that is, background removed), and one object (gazebo) was slightly modified using GNU image manipulation software v.2.1 (http://www.gimp.org) to increase its mirror symmetry. We grouped the pictures into three different sets of three, always combining objects with different aspect ratios (width/height; see the example set in Fig. 3a). Each participant was assigned one of these sets, with each set being used similarly often across the participant sample (two sets were used 18 times and one set 19 times). As auditory cue stimuli, we prepared recordings of the words 'one', 'two' and 'thanks' spoken by a female lab member. The recordings were time-compressed to a common length of 350 ms using a pitch-preserving algorithm provided in Audacity v.2.3.0 (GNU software; https://www.audacityteam.org/).

Each trial started with a fixation dot (8 × 8 px, corresponding to a 0.17 × 0.17° visual angle) displayed at the centre of the screen for 500–1,000 ms (randomly varied), followed by sequential presentation of two objects, each in a random orientation (see below). Each stimulus was displayed for 500 ms (display size ~6.5° visual angle, see Fig. 1c) followed by a 500 ms blank screen. After this, an auditory retro-cue ('one' or 'two', 350 ms) indicated which of the two stimulus orientations was to be reported after a delay (Delay 1, 3,500 ms) in the upcoming memory test (Test 1). Test 1 started with the cued object reappearing on display, but with its previous orientation changed by ±6.43°. Participants were asked to indicate by means of key press (2-AFC) whether the object would need to be rotated clockwise or anticlockwise (right or left arrow key) to match its memorized orientation. On key press, the object rotated accordingly (by 6.43°), followed by a written feedback message ('correct' or 'incorrect') displayed in the upper part of the screen (500 ms). After another 500 ms, in half the trials, an auditory message ('thanks', 350 ms) signalled the end of the trial. In the other half of the trials (randomly varied), a second auditory retro-cue (Cue 2) was presented (for example, 'two', if the first retro-cue was 'one'), indicating that the thus-far-untested stimulus orientation would still need to be reported. In these trials, another delay period ensued (Delay 2, 2,500 ms), and participants' memory for the second-cued stimulus was tested (Test 2), using the same procedure as before for the first-cued stimulus in Test 1. Each participant performed 16 blocks of 32 trials, for a total of 512 trials (265 of which included a Test 2).

Stimulus presentation was pseudorandom across trials, with the following restrictions: (1) each pairing of objects from the participant's object set occurred equally often, (2) each object was equally often presented first (as Stimulus 1) and second (as Stimulus 2), and (3) Stimulus 1 and Stimulus 2 were equally often cued for Test 1. The orientations of the two objects on each trial were drawn randomly and independently from 16 equidistant values (11.25° to 348.75° in steps of 22.5°), which excluded the cardinal axes (0°, 90°, 180° and 270°).

The experiment was run using Psychophysics Toolbox v.3 (ref. 72) with the included Eyelink Toolbox[73] in MATLAB 2017a (MathWorks). The visual stimuli were presented on a 60 × 34 cm screen with a 2,560 × 1,440 px resolution and a frame rate of 60 Hz. The auditory cue words were presented through desktop loudspeakers (Harman Kardon KH206). To minimize head motion, participants performed the experiment with their head positioned on a chin rest with a viewing distance of ~62 cm from the screen. Gaze position was monitored and recorded throughout the experiment at a sampling rate of 500 Hz using a desktop-mounted EyeLink 1000 eye-tracker (SR Research), with file and link/analogue filters set to 'EXTRA' and 'STD', respectively.

Participants were instructed to constantly keep their gaze on the fixation dot, which was displayed throughout the entire trial except for the test and feedback periods. Whenever a participant's gaze deviated more than 71 px (1.53° visual angle) from the centre of the fixation dot either before object presentation or for longer than 500 ms during any of the two delay periods, a warning message ('Fixate') was displayed at the centre of the screen. This occurred during less than 15% (mean: 13.03%) of the trial epoch on average.

### Behavioural modelling

To model participants' behavioural memory reports (2-AFC), we used a geometrical approach similar to that used in our eye-tracking analyses (see below). We first defined three prototypical geometries: (1) an unbiased 'circle' model ($M_{circle}$) corresponding to the memory items' 16 original orientations (Fig. 4a, middle), (2) a cardinal repulsion model ($M_{repulsion}$) that shifts the 16 orientations to the nearest diagonal orientation (that is, 45°, 135° 225° or 315°; see Fig. 4a, rightmost) and (3) a cardinal attraction model ($M_{attraction}$) that shifts them to the nearest cardinal orientation (that is, 0°, 90°, 180° or 270°; see Fig. 4a, leftmost). The continuum from attraction to repulsion was formalized with a mixture parameter $B$ (ranging from −1 to 1), which blends the circle model with the repulsion model for $B \geq 0$

$$M_{mix} = BM_{repulsion} + (1 - B)M_{circle}$$

and with the attraction model for $B < 0$

$$M_{mix} = -BM_{attraction} + (1 + B)M_{circle}$$

Figure 4a illustrates the resulting model continuum from $B = -1$ (maximal attraction) over $B = 0$ (unbiased) to $B = 1$ (maximal repulsion). To simulate memory reports (clockwise or anticlockwise) for each trial, we computed the angular difference $d$ between the orientation modelled in $M_{mix}$ and the probe orientation displayed at test and transformed it into a probability of making a 'clockwise' response using a logistic choice function

$$P_{clockwise} = \frac{1}{1 + \exp(-d/s)}$$

where $s$ is a noise parameter that relates inversely to memory strength or precision (see also ref. 74). For completeness, our model also allowed for greater memory precision near the cardinal axes (a so-called oblique effect[18,75]). This was implemented by a further parameter $c$, which up- or downregulated noise $s$ for those eight orientations in the stimulus set that were near the cardinal axes (Fig. 4a, middle) relative to the remaining eight orientations that were nearer the diagonal axes

$$s_{\text{near-cardinal}} = \exp(c)s_{\text{near-diagonal}}$$

where values of $c < 0$ would indicate relatively greater precision (lower noise) near the cardinal axes (that is, an oblique effect). The model was fitted to the memory reports of each participant individually using exhaustive gridsearch ($B$, −1...1; $s$, 0...1; $c$, −0.5...0.5; with a step size of 0.01 for each parameter) and least squares to identify the best-fitting parameter values.

Although our analysis focused on bias ($B$), we note for completeness that we also observed values of $c$ significantly smaller than 0 (mean across conditions: $c = -0.292$, $t(40) = -10.159$, $P < .001$, $d = -1.586$, 95% CI (−0.350, −0.234), $t$-test against 0): that is, an oblique effect, which replicates and extends previous work[18]. The strength of this oblique effect tended to decrease with mnemonic distance ($t(40) = 2.286$, $d = 0.357$, $P = 0.028$; $t$-test of linear slope against zero) (see ref. [18] for related findings).

### Eye-tracking analysis

The eye-tracking data were only minimally preprocessed. The data from each participant were zero-centred (using the overall mean over all trials), and data points with a Euclidean distance larger than 100 px (corresponding to a 2.17° visual angle) from the zero-centre were excluded from analysis (Fig. 1c and Fig. 5 show data before this exclusion). We analysed the data in two epochs of interest, one time-locked to Stimulus 1 onset (from −500 ms until the onset of Test 1 at 5,850 ms) and the other time-locked to Cue 2 onset (from −500 ms until the onset of Test 2 at 2,850 ms). After artefact exclusion, on average 97.87% (s.d. = 1.50%, first epoch) and 95.14% (s.d. = 3.67%, second epoch) of the data remained for analysis.

**Representational similarity analysis.** RSA of the gaze-position data was performed separately for each participant using a single-trial approach. For each trial, we first obtained the trial average for each of the 16 orientations while leaving out the current trial. We then computed at each time point the 16 Euclidean distances between the gaze position in the current trial and the trial averages formed from the remaining data. This yielded a representational dissimilarity vector (RDV) of the distances between the (single-trial) gaze associated with the orientation in the current trial and the (trial-averaged) gaze associated with each of the 16 orientations (Fig. 2b). To examine orientation encoding, we computed at each time point and for each trial the Pearson correlation ($\rho$) between the empirical RDV and the theoretical RDV predicted under a model of orientation encoding (see below) for the orientation on the current trial. When averaged over trials (and hence also across orientations), the procedure yields a leave-one-out cross-validated time course of orientation encoding, similar to more conventional RSA approaches with trial averages. However, the single-trial approach additionally retains the trial-by-trial variability in orientation encoding (Fig. 2b, right, and Fig. 2e).

To examine orientation encoding within and between objects (Fig. 3a), we used the same approach but obtained the 16 trial averages separately for each of the three different objects in the participant's stimulus set. This yielded three empirical RDVs per trial (one within and two between objects) that were independently correlated (Pearson's $\rho$) with the model RDV. The two between-objects correlations were then averaged.

All RSA results were obtained individually for each participant and examined statistically on the group level. We used cluster-based permutation testing[76], where we first identified clusters of consecutive samples that showed an effect with $P_{\text{sample}} < 0.05$ (uncorrected) and calculated the sum of $t$-values in a cluster as its test statistic. We then estimated the probability $P_{\text{cluster}}$ that a cluster with a larger test statistic would emerge by chance, on the basis of 20,000 iterations where the individual participant effects were randomly sign-flipped. Unless otherwise specified, all reported statistical tests were two-sided.

**Model geometries.** Our basic orientation model was a perfect circle geometry (Fig. 2a, left), where the model RDVs reflected the pairwise

Euclidean distances between 16 evenly spaced points on a circle (Fig. 2a, right; note that each line in the distance matrix corresponds to the model RDV for a given stimulus orientation (Fig. 2b)). The geometry of this model corresponds to our behavioural analysis model with $B = 0$ (that is, $M_{\text{circle}}$, unbiased). To examine bias in the gaze patterns (Fig. 4), we used the Euclidean distance structures associated with our maximally biased models with $B = -1$ ($M_{\text{attraction}}$) and $B = 1$ ($M_{\text{repulsion}}$), respectively. Comparing these two extreme models (which both have a square geometry) yields an estimate of the extent to which the gaze patterns were repulsively or attractively biased (Results). Note that the distance structures expected under the three different models ($B = 0$, $B = 1$ and $B = -1$) correlate with each other ($r = 0.77$ and 0.34). We thus did not expect very large differences in their fit of the data and report the results with a more liberal statistical threshold ($P_{\text{cluster}} < 0.05$).

**Microsaccade detection.** For complementary analysis of microsaccades (Fig. 5), we used a velocity-based detection algorithm established in previous work[77–79]. In brief, the gaze-position data were transformed into a velocity time course by calculating the Euclidean distances between consecutive samples and smoothing with a 7 ms Gaussian kernel. Saccade onsets and endpoints were inferred from when the gaze velocity exceeded a trial-specific threshold (5 times the median velocity in the trial) and when it returned to below threshold, with a minimum interval of 100 ms between successively detected saccades.

### Reporting summary

Further information on the research design is available in the Nature Portfolio Reporting Summary linked to this article.

## Data availability

The data that support the findings of this study are available at https://gin.g-node.org/lindedomingo/mpib_memoreye.

## Code availability

The experiment and analysis code is available at https://gin.g-node.org/lindedomingo/mpib_memoreye.

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

## Acknowledgements

We thank I. Padezhki, C. Wicharz, J. Hebisch, A. Faschinger, G. Inciuraite and A. Anouk Bielefeldt for their help with data collection and J. Wäscher for participant recruitment. We also thank J. Hebisch for recording the auditory stimuli, M. Rolfs for helpful comments and discussion, R. Hertwig for general support and T. Graham for editorial assistance. Part of this work was conducted at the Max Planck Dahlem Campus of Cognition of the Max Planck Institute for Human Development, Berlin, Germany. This research was supported by European Research Council Consolidator Grant ERC-2020-COG-101000972 (B.S.) and by DFG grant SP 1510/7-1 (B.S.). J.L.-D. received support from Ramon-y-Cajal fellowship RYC2021-033940-I by the Spanish Ministry of Science and Innovation. The funders had no role in the study design, data collection and analysis, decision to publish or preparation of the manuscript.

## Author contributions

J.L.-D. was responsible for data curation, formal analysis, investigation, validation and visualization. B.S. was responsible for funding acquisition, methodology, resources and supervision. Both authors contributed to study conceptualization, project administration, and writing, reviewing and editing the manuscript.

## Funding

## Competing interests

The authors declare no competing interests.

## Additional information

**Correspondence and requests for materials** should be addressed to Juan Linde-Domingo or Bernhard Spitzer.

# Reporting Summary

## Statistics

For all statistical analyses, confirm that the following items are present in the figure legend, table legend, main text, or Methods section.

| n/a | Confirmed | |
|---|---|---|
| ☐ | ☒ | The exact sample size (n) for each experimental group/condition, given as a discrete number and unit of measurement |
| ☐ | ☒ | A statement on whether measurements were taken from distinct samples or whether the same sample was measured repeatedly |
| ☐ | ☒ | The statistical test(s) used AND whether they are one- or two-sided *Only common tests should be described solely by name; describe more complex techniques in the Methods section.* |
| ☒ | ☐ | A description of all covariates tested |
| ☐ | ☒ | A description of any assumptions or corrections, such as tests of normality and adjustment for multiple comparisons |
| ☐ | ☒ | A full description of the statistical parameters including central tendency (e.g. means) or other basic estimates (e.g. regression coefficient) AND variation (e.g. standard deviation) or associated estimates of uncertainty (e.g. confidence intervals) |
| ☐ | ☒ | For null hypothesis testing, the test statistic (e.g. $F$, $t$, $r$) with confidence intervals, effect sizes, degrees of freedom and $P$ value noted *Give P values as exact values whenever suitable.* |
| ☒ | ☐ | For Bayesian analysis, information on the choice of priors and Markov chain Monte Carlo settings |
| ☒ | ☐ | For hierarchical and complex designs, identification of the appropriate level for tests and full reporting of outcomes |
| ☐ | ☒ | Estimates of effect sizes (e.g. Cohen's $d$, Pearson's $r$), indicating how they were calculated |

*Our web collection on statistics for biologists contains articles on many of the points above.*

## Software and code

Policy information about availability of computer code

| Data collection | The experiment was run using Psychophysics Toolbox Version 3 (PTB; Brainard & Vision, 1997) and its incorporated Eyelink Toolbox version for PTB Version 3 (Cornelissen et al., 2002) in MATLAB 2017a (MathWorks). |
|---|---|
| Data analysis | All analyses were run using MATLAB 2018a (MathWorks). |

For manuscripts utilizing custom algorithms or software that are central to the research but not yet described in published literature, software must be made available to editors and reviewers. We strongly encourage code deposition in a community repository (e.g. GitHub). See the Nature Portfolio guidelines for submitting code & software for further information.

## Data

Policy information about availability of data

All manuscripts must include a data availability statement. This statement should provide the following information, where applicable:
- Accession codes, unique identifiers, or web links for publicly available datasets
- A description of any restrictions on data availability
- For clinical datasets or third party data, please ensure that the statement adheres to our policy

The data that support this study are available at https://gin.g-node.org/lindedomingo/mpib_memoreye

## Research involving human participants, their data, or biological material

Policy information about studies with <u>human participants or human data</u>. See also policy information about <u>sex, gender (identity/presentation), and sexual orientation</u> and <u>race, ethnicity and racism</u>.

| | |
|---|---|
| Reporting on sex and gender | Fifty-five participants (31 female, 24 male) took part in the experiment. |
| Reporting on race, ethnicity, or other socially relevant groupings | N/A -- no such data were collected (besides age and sex), and they played no role in participant recruitment |
| Population characteristics | Participants (young adults of any gender) were recruited from the general population of the city of Berlin (Germany) and surrounding areas with an interest in participating in scientific studies, in a age range of 18-35 years (mean age 26.95 ± 3.98 years). |
| Recruitment | We recruited young adult participants of any sex or gender. Potential participants were informed about receiving a compensation of €10 per hour plus a bonus based on task performance (€5 bonus if four out of five randomly selected memory reports were correct). Written informed consent was obtained from all participants prior to participation. |
| Ethics oversight | Deutsche Gesellschaft für Psychologie (DGPs), Bonn, Germany |

Note that full information on the approval of the study protocol must also be provided in the manuscript.

# Field-specific reporting

Please select the one below that is the best fit for your research. If you are not sure, read the appropriate sections before making your selection.

☒ Life sciences  ☐ Behavioural & social sciences  ☐ Ecological, evolutionary & environmental sciences

For a reference copy of the document with all sections, see nature.com/documents/nr-reporting-summary-flat.pdf

# Life sciences study design

All studies must disclose on these points even when the disclosure is negative.

| | |
|---|---|
| Sample size | Pilot experiments in our lab with similar stimulus materials showed that stimulus orientation was robustly reflected in gaze position in a sample of n = 20 participants during a simple WM maintenance period. Since the present experiment additionally included a second maintenance period that occurred only in half of the trials, we approximately doubled the sample size for the present study (n = 55 participants were recruited, of which n = 41 remained for analysis, see below). While no formal power analysis was conducted, post-hoc Bayesian Anlayses confirmed that the sample size was sufficient both to detect evidence for the presence and for the absence of an effect (e.g., Fig. 3) |
| Data exclusions | Two participants (both wearing glasses) were excluded due to difficulties in acquiring a stable eye-tracking signal, and one participant was excluded because they reported feeling unwell during the experimental session. Of the remaining participants, we excluded n = 9 for failing to perform above chance level in each of the two memory tests (p < 0.05, Binomial test against 50% correct responses). Finally, after preprocessing the eye-tracking data, we excluded n = 2 participants for whom more than 15% of the data had to be rejected due to blinks and other recording artifacts. After this, n = 41 participants remained for analysis. |
| Replication | N/A - The study includes no direct replication attempt of a previous finding. Aspects of our results that conceptually align with earlier findings are described as such in the manuscript text. |
| Randomization | The experiment was a within-subjects design without distinct experimental groups. The within-subjects task conditions were randomized as stated in Methods |
| Blinding | N/A -- The experiment was a within-subjects design where each participant performed the same variants of a behavioral task (blinding not applicable) |

# Reporting for specific materials, systems and methods

We require information from authors about some types of materials, experimental systems and methods used in many studies. Here, indicate whether each material, system or method listed is relevant to your study. If you are not sure if a list item applies to your research, read the appropriate section before selecting a response.

## Materials & experimental systems

| n/a | Involved in the study |
|-----|----------------------|
| ☒ | Antibodies |
| ☒ | Eukaryotic cell lines |
| ☒ | Palaeontology and archaeology |
| ☒ | Animals and other organisms |
| ☒ | Clinical data |
| ☒ | Dual use research of concern |
| ☒ | Plants |

## Methods

| n/a | Involved in the study |
|-----|----------------------|
| ☒ | ChIP-seq |
| ☒ | Flow cytometry |
| ☒ | MRI-based neuroimaging |

