## [Peer Review File · Nature Human Behaviour]

Peer Review Information

Journal: Nature Human Behaviour

Manuscript Title: Geometry of visuospatial working memory information in miniature gaze patterns

Corresponding author name(s): Juan Linde-Domingo and Bernhard Spitzer

Reviewer Comments & Decisions:

Decision Letter, initial version:

13th March 2023

Dear Dr Linde-Domingo,

Thank you once again for your manuscript, entitled "Geometry of visual working memory information in human gaze patterns", and for your patience during the peer review process.

Your article has now been evaluated by 3 referees. You will see from their comments copied below that, although they find your work of potential interest, they have raised quite substantial concerns. In light of these comments, we cannot accept the manuscript for publication, but would be interested in considering a revised version if you are willing and able to fully address reviewer and editorial concerns.

We hope you will find the referees' comments useful as you decide how to proceed. If you wish to submit a substantially revised manuscript, please bear in mind that we will be reluctant to approach the referees again in the absence of major revisions. We are committed to providing a fair and constructive peer-review process. Do not hesitate to contact us if there are specific requests from the reviewers that you believe are technically impossible or unlikely to yield a meaningful outcome.

To guide the scope of the revisions, the editors discuss the referee reports in detail within the team, including with the chief editor, with a view to (1) identifying key priorities that should be addressed in revision and (2) overruling referee requests that are deemed beyond the scope of the current study. We believe that new experimental data will be required to address some of the concerns raised by the reviewers.

Specifically, we ask you to perform an additional experiment including a control task to rule out that the abstraction effects are not due to visual masking or passing of time (Reviewer 1). We leave at your own discretion whether to perform also the generalization experiment suggested by Reviewer 3 to strengthen the conclusions beyond the current task (e.g., color instead of orientation detection).

Please do not hesitate to get in touch if you would like to discuss these issues further.

If you wish to submit a suitably revised manuscript, we would hope to receive it within 4 months. I would be grateful if you could contact us as soon as possible if you foresee difficulties with meeting this target resubmission date.

- Include a "Response to the editors and reviewers" document detailing, point-by-point, how you addressed each editor and referee comment. If no action was taken to address a point, you must provide a compelling argument. When formatting this document, please respond to each reviewer comment individually, including the full text of the reviewer comment verbatim followed by your response to the individual point. This response will be used by the editors to evaluate your revision and sent back to the reviewers along with the revised manuscript.
- Highlight all changes made to your manuscript or provide us with a version that tracks changes.

[REDACTED]

Thank you for the opportunity to review your work. Please do not hesitate to contact me if you have any questions or would like to discuss the required revisions further.

Sincerely,

Giacomo Ariani
Editor
Nature Human Behaviour

Reviewer expertise:

Reviewer #1: Visual working memory, eye-tracking, RSA

Reviewer #2: Visual attention, microsaccades

Reviewer #3: Working memory, eye movements

REVIEWER COMMENTS:

Reviewer #1:

Remarks to the Author:

The authors investigated whether fixational gaze behaviour can be used to track visual working memory content, as well as changes in priority states and abstraction in working memory, across time. They successfully used representational-similarity analysis (RSA) to track stimulus orientation information when working memory contents were encoded, attention, and re-attended. While systematic biases in gaze behaviour by memorised stimulus orientation have already been reported in several prior publications, the authors introduce a sensitive RSA approach for analysing such biases, and extend prior studies by also tracking the attentional status and abstraction of memory items across time.

Overall, the article is well written and provides an elegant demonstration of the tight link between eye-movements and visual working memory. At the same time, a central open question remains to what extent gaze allows to truly infer something about the quality of representation as retained by the brain during working memory (though a similar reservation can probably be applied to other measures, such as fMRI or EEG decoding traces). I elaborate below.

Major

Making inferences on representational format/abstraction of working memory representations from patterns of gaze behaviour remains a big leap. For example, in the discussion it is suggested that the central findings in Figure 3 (comparing within vs. between-object decoding) show how visual memories become less concrete (abstracted) with time. While these results show that the gaze patterns become less object-specific, I am not sure we can conclude that therefore also the visual representation has become less concrete/detailed. The brain may retain a visual memory with the same level of detail, but simply no longer require eye movements for scanning this level of detail, as time passes.

In addition to the above, this “abstraction” effect appears particularly pronounced for stimulus 1, starting as soon as stimulus 2 is presented. I wonder to what extent this may be driven by visual masking (which may or may not also affect the visual representation). Indeed, when considering stimulus 2, that is not followed by another visual stimulus, the object specificity seems to last much longer.

A convincing way to demonstrate that the abstraction effect is not due to visual masking and/or mere passage of time, but genuinely reflects the demands of the working-memory task, would be to manipulate the task. For example, imagine the task would not involve detecting a difference in orientation, but a difference in the visual details of the object. Now, the abstraction effect toward an orientation-specific, object-invariant code should be there in the orientation task, but not in the object-detail task, for which such abstraction would be undesirable.

Line 285 states how the orientation coding in gaze has “become fully object-independent.” However,

this reflects a null effect in the comparison between within-object and between-object decoding. On the basis of a null effect one cannot conclude full independence.

Minor

Currently, report 2 only occurs in 50% of trials, whereas report 1 occurs in 100% of trials. In this way, unattended is conflated with less relevant, because the chance of being tested for the unattended item is only 50%. This may explain why people are worse – not because the item had been unattended, but because it was deemed less relevant for the task and therefore possibly strategically dropped / devoted less resources to.

When comparing the repulsion vs. attraction RSA model-correlations with gaze, the author find that the repulsion model works better. However, the best model always seems to be the unbiased model, which is the in-between model. Leaving the unbiased model out of the comparison appears odd, and I am not sure how to interpret the repulsion vs. attraction comparison, if the middle-ground-model always works best.

Also, the authors never included cardinal orientations in their stimulus set. Might this contribute to the tendency for participants to repulse from cardinal orientations, strategically?

The authors discuss their results in the context of related literature on microsaccades. While I agree this is sensible, the authors did minimal processing of their eye data, and did not quantify saccades. Fixational gaze behaviour is not only driven by microsaccades, but also by drifts and tremor. This may be acknowledged and/or could be quantified.

Fig 3: it might be nice to plot the object-specific and object-independent effects as their own time courses, to see the transition in time. Currently the reader has to imagine how the difference/average looks over time.

The first sentence of the abstract appears suboptimal. This currently reads as if this article is about neural decoding confounds. Rather, it is about representational formats in working memory, and how to track them through gaze.

There are some formatting issues, such as an empty formula and occasional symbols that seem out of place.

Were eye-movements the primary outcome measure for this study, or was this study designed as an fMRI study, or alike? This may be good to disclose.

In the introduction it is claimed how 1-item studies are typical in working memory research. I am not sure this is a fair depiction. For instance, what about all the studies on working memory capacity?

In the first paragraph on the eye tracking results, the authors could make more explicit that this specific aspect (gaze biased by memory orientation) is consistent with several recent other studies that showed similar findings (the authors are aware of these studies).

Reviewer #2:

Remarks to the Author:

Review on Linde-Domingo & Spitzer: Geometry of visual working memory information in human gaze patterns

Overall evaluation: I am impressed by reading this manuscript reporting remarkable findings on the information that can be obtained from miniature eye movements during active fixation. The experimental procedures and the analyses seem sound, the writing is clear and succinct, and the conclusions are reasonable. I have only few questions and remarks. Congrats to the authors.

Comments:

1) It is difficult to understand how the authors could reconstructed the full angle of 360 degrees (2π), since there obviously an ambiguity in the mapping from the rotated to the original object. Therefore, I would have expected an orientation plot with re-rotation angles over a range of 180 degrees (π). Could you please clarify the explanations for Fig. 1 b & c?

2) I am not convinced how the effect is generated. For example, when presented real-world object with a complex geometry as in the current experiment, it is hard to predict where the "visual center of gravity" is (where is the mean fixation position). Thus, the effect could be explained by tiny shifts in such an average fixation position from the presented location. Data to investigate such a hypothesis are there: What was the average fixation position during "Stimulus 1" and "Stimulus 2" epochs as a function of rotation angle? Did the authors observe any biases here? If yes, then an explanation of the effect could be independent of the orientation, but merely be remembering the original fixation bias during the perceptual encoding.

3) In the discussion, the authors relate the current findings to previous work on microsaccades and attentional effect. However, there is no reason to speculate on the microsaccadic effects, given that the data are recorded and microsaccadic activity could be analyzed to clarify the picture. This would be a useful extension of the current analyses. This point is not critical for a potential revision, but could give a more complete view into the gaze dynamics during the different epochs of the experiment.

4) In my PDF, some of the equations and formula characters were not readable (mainly due to missing characters). Therefore, I could not check their correctness.

EOF

Reviewer #3:

Remarks to the Author:

In the manuscript NATHUMBEHAV-23010084 entitled „Geometry of visual working memory information in human gaze patterns" the authors describe the results of one experiment in which they analyze miniature eye movements (defined as saccadic eye movements smaller than 1° of visual angle) during a dual retro-cue task testing orientations of rotated objects. Their results show that miniature eye movements contain rich information on the representational format in visual working memory. During

stimulus presentation, eye movement patterns were in line with orientation encoding and changed to more object-independent formats during cued maintenance. The more attention had been withdrawn from one stimulus, the more abstract the representational format. Even more, during periods of unattended storage, categorical reporting biases emerged.

Using eye movements to study visual working memory is an advancing field. The manuscript provides an excellent example of the usefulness of studying eye movements to investigate memory processes (i.e. the nature and time course of representation in visual working memory) and seems suitable for publication in Nature Human Behavior when appropriately addressing my concerns listed below.

Overall, the presented experiment is carefully conducted, and the analyses are of highest quality. I wonder a bit why they aggregated the data prior to the frequentist analyses and did not use hierarchical analyses that would allow to account for interindividual variability in the strength of the observed effects. I had some problems understanding the calculation of rho and pcluster. I would wish the authors to describe the calculation of these analyses more carefully. Maybe, I missed an important piece as not all formulas were displayed correctly in the available pdf (p. 19, l. 642, 651, 653 onwards).

Eye movements are guided by various top-down and bottom-up factors. Consequently, eye movements and especially miniature eye movements are sometimes very noisy measures of human behavior (even in such a controlled experimental environment as in the present experiment). In addition, the authors aim to establish miniature eye movements and their analyses approach as a tool to get insights into visual working memory. To be more convincing that the results would replicate and generalize to other task typically used to study visual working memory, I would suggest the authors show the same set of effects in a new experiment. For instance, a typical visual working memory task is to remember the color of squares presented at different screen locations. If colors were arranged in a circle, the same analysis strategy may be applicable.

The figures are very well designed, however in some cases, they are too small. For instance, the polar plot in Figure 4b is very small. Furthermore, I miss the description of the error bands in the plots with timeline data (e.g., Figure 4c, Figure 3c). I cannot see purple markers below the plots in Figure 4c. Some plots have a red line at the bottom.

I do not quite agree with the criticism that the results are at odds with the idea of a single-item focus of attention. Only one item can be in the focus of attention, however several items can be in an activated state in working memory and held active with maintenance strategies. The data is highly aggregated but on the trial level one can only look at one position at a time. Thus, the authors should be more specific at how their results are at odds with these current theories of working memory.

Author Rebuttal to Initial comments

We are grateful for the reviewers' thorough and constructive evaluation of our manuscript.

Reviewer #1:

Remarks to the Author:

The authors investigated whether fixational gaze behaviour can be used to track visual working memory content, as well as changes in priority states and abstraction in working memory, across time. They successfully used representational-similarity analysis (RSA) to track stimulus orientation information when working memory contents were encoded, attention, and re-attended. While systematic biases in gaze behaviour by memorised stimulus orientation have already been reported in several prior publications, the authors introduce a sensitive RSA approach for analysing such biases, and extend prior studies by also tracking the attentional status and abstraction of memory items across time.

Overall, the article is well written and provides an elegant demonstration of the tight link between eye-movements and visual working memory. At the same time, a central open question remains to what extent gaze allows to truly infer something about the quality of representation as retained by the brain during working memory (though a similar reservation can probably be applied to other measures, such as fMRI or EEG decoding traces). I elaborate below.

We thank the reviewer for this positively balanced assessment of our study.

Major

Making inferences on representational format/abstraction of working memory representations from patterns of gaze behaviour remains a big leap. For example, in the discussion it is suggested that the central findings in Figure 3 (comparing within vs. between-object decoding) show how visual memories become less concrete (abstracted) with time. While these results show that the gaze patterns become less object-specific, I am not sure we can conclude that therefore also the visual representation has become less concrete/detailed. The brain may retain a visual memory with the same level of detail, but simply no longer require eye movements for scanning this level of detail, as time passes.

The reviewer emphasises an important point, which indeed applies similarly to other measures such as fMRI, M/EEG, or LFPs as well. We are aware of the general limitations of correlative evidence, and we carefully checked our writing to not suggest we would be measuring visual representations directly, let alone comprehensively, when examining gaze patterns. It is indeed possible that the brain may retain information (e.g., visual details) in ways that may not register in ocular activity. The same reservations would of course also hold for any other (e.g., neural) measures in contemporary WM research. Please see also the 2nd-last paragraph in our revised Discussion, where we considered general limitations

inherent in the eye-tracking approach, and which we extended in response to the reviewer's comment.

In addition to the above, this “abstraction” effect appears particularly pronounced for stimulus 1, starting as soon as stimulus 2 is presented. I wonder to what extent this may be driven by visual masking (which may or may not also affect the visual representation). Indeed, when considering stimulus 2, that is not followed by another visual stimulus, the object specificity seems to last much longer.

We understand that this could be a concern for the reviewer but we believe their comment may have overlooked a small but important detail in the precise timing of our results in Fig. 3c. The transition to object-independent encoding of Stimulus 1's orientation did not start as soon as stimulus 2 was presented. Instead, the effect occurred only approx. 300-500 ms after Stimulus 2's onset (Fig 3c, *upper*), at the exact same time at which Stimulus 2's orientation was itself **encoded** in gaze (Fig. 3c, *lower*; see also p. 7, 2nd paragraph).

With this important clarification in mind, we think it is less likely that the effect would merely reflect visual masking. We would argue that for the orientation of Stimulus 2 to be *encoded in gaze*, the second object must either already have been recognized (and its orientation perceived), or at least its salient features must have already attracted the observer's visual attention. We therefore argue in favour of our original interpretation that the rapid decline of within-objects encoding for Stimulus 1 is likely linked to the attentional orienting towards Stimulus 2 (and thus, to the partial withdrawal of attention from Stimulus 1) in the exact same time window, rather than only reflecting an automatic response to (any) visual input.

The reviewer is correct that the abstraction effect was more pronounced for Stimulus 1 than for Stimulus 2, which is part of our central findings (see Discussion). However, the overall tendency (object-specific shortly after stimulus presentation, more object-independent during sustained WM maintenance) was evident also for Stimulus 2 (Fig. 3c, bottom panel), just not as strongly as for Stimulus 1. Stimulus 2 was not followed by any visual input (Fig. 1a), which adds to the evidence against visual masking as the primary cause of the increasingly object-independent gaze patterns during WM processing.

A convincing way to demonstrate that the abstraction effect is not due to visual masking and/or mere passage of time, but genuinely reflects the demands of the working-memory task, would be to manipulate the task. For example, imagine the task would not involve detecting a difference in

orientation, but a difference in the visual details of the object. Now, the abstraction effect toward an orientation-specific, object-invariant code should be there in the orientation task, but not in the object-detail task, for which such abstraction would be undesirable.

Before addressing these suggestions in detail, we hope our previous reply has already ameliorated the reviewer's primary concern that the results in Fig. 3 would only be due to visual masking. We argue that both the exact timing of the effect for Stimulus 1 and the similar overall tendency for Stimulus 2 (which was unmasked) speak against such alternative interpretation.

The reviewer was further concerned that the effects may only result from the mere passage of time. However, inspection of Fig. 3c shows that the patterns (within vs. between) were in fact very stable throughout the Delay 1 period of 3500 ms, which is a fairly long time in WM, and this was true for either Stimulus (1 and 2). In the revision, we confirmed this temporal stability using Bayesian testing (BF_{10}), which yielded evidence in favour of the null hypothesis that the within-between difference did not change over time, neither for Stimulus 1 nor Stimulus 2 ($BF_{10} = 0.168$ and $BF_{10} = 0.208$; Bayesian t-tests comparing the within-between difference between the first and the last 1000 ms of Delay 1). We thus have little reason to assume that object-independence increased regardless of WM demands, simply with the passage of time. More generally, we would not expect to observe a sustained, and even ramping-up encoding of stimulus orientations in gaze at all, if it was not for the WM demands of our task: please note that this sustained, ramping-up encoding was exclusively evident for the *retrospectively cued* WM information (Fig. 2-4).

Notwithstanding these important clarifications, we thoroughly considered also the reviewer's suggestion of performing another experiment which would require participants to detect a difference in the objects' visual details, rather than judging a difference in orientation. We agree with the reviewer that in such an alternative task, an orientation-specific, object-invariant code would likely not be helpful. Based on the accumulating existent literature, we would even go further and predict that in such a task, potential stimulus-dependent gaze patterns would not even directly reflect *orientation*, but rather the *spatial location* of the stimulus feature(s) in question. To elaborate, we considered the following hypothetical variants of such task:

First, if participants would not be informed beforehand which specific detail(s) of the sample stimuli might change at test, the gaze patterns (during encoding and maintenance alike) can be expected to become very complex. Participants may mentally "scan" the object parts for a comprehensive encoding (and possibly also maintenance) of its many visual details. It is

unclear a priori, whether such complex patterns would even systematically ‘rotate’ with the stimulus orientation (around 360°; see Fig.1) – which, however, would be a prerequisite for our RSA approach to disclose any orientation encoding (be it object-specific or object-independent) in the first place.

Alternatively, one could inform participants beforehand about which specific object detail may change (say, the door on bottom of our lighthouse stimulus). However, then the task would reduce to maintaining a single piece of stimulus information that occurred in a certain spatial *location*. Many previous WM studies have already shown sustained micro-saccading to the spatial location of to-be-maintained stimulus features (De Vries & Van Ede, 2023; van Ede et al., 2019). The outcome of such a hypothetical experiment can thus be anticipated based on already existing work. In our RSA analysis of object *orientation* (Fig 2-3), the expected gaze pattern would very likely masquerade as an ‘object-specific’ encoding of orientation (simply because the spatial location of the object-specific target feature in question would vary circularly with the object’s orientation). However, we would argue that this would only reflect a (known) gaze bias to the instructed target feature’s location, and not an *intrinsically* object-specific encoding of orientation that would be comparable to our present study.

Yet another alternative would be to ask participants to detect e.g., a change in the object’s overall colour (e.g., the red tone of the boombox in Fig. 1a), its size, or similar “global” features that are orientation-invariant by definition. For such tasks, stimulus orientation would be entirely task-irrelevant, and we would expect no orientation-dependent gaze patterns in the first place, potentially not even during perceptual encoding.

To summarize our concerns about the suggested alternative experiment, we believe it can already be inferred from existing work that one would be unlikely to observe an object-independent gaze signature of object orientation in such an experiment. We would rather expect to either observe no clear orientation encoding in gaze at all, or instead a signature of the target feature(s)’ *location*, which has already been shown in earlier studies, e.g., van Ede et al., 2019), and which would only masquerade as object-specific “orientation” encoding in our analysis framework. We therefore do not think that the anticipated results from such an experiment would further clarify the nature of the effects observed in our present study, where importantly, the emergence of object-specific vs object-independent gaze patterns was purely intrinsic.

We believe that a key strength of our paradigm is that the task could a priori be solved with a concrete/pictorial ‘visual’ memory, just as it could be solved by remembering a high-level abstraction of the task-relevant information (see Introduction and motivation for our study).

Our analysis was tailored to this very approach to yield evidence for (or against) an *intrinsic* focusing on concrete visual aspects vs. high-level abstractions. The same would not hold, we believe, in a task that essentially *requires* remembering visual details to perform the task. Applying our tailored approach to a domain it was not designed for (i.e. memory for visual details), in our view, would lead to problems in interpretation and comparability.

All these things considered, we believe that such a hypothetical experiment, albeit potentially interesting, would not directly speak to the reviewer's original concern that part of our results (Fig. 3) might have been driven by visual masking, or by the mere passage of time. We hope our earlier replies and additional analyses (see below) have ameliorated these original concerns.

To conclude, we would agree that our current study, as the first of its kind, leaves open questions for future work, including whether abstraction/generalisation effects would also be promoted e.g., by task-irrelevant distractor inputs, and/or with higher memory loads. However, in our view, these are interesting research questions in their own right, and best to be addressed in dedicated future studies.

Line 285 states how the orientation coding in gaze has "become fully object-independent." However, this reflects a null effect in the comparison between within-object and between-object decoding. On the basis of a null effect one cannot conclude full independence.

We agree, and we added new analyses (Bayes Factors, BF, on p. 7) and figure panels (Fig. 3e-f) which substantiate this interpretation. In Delay 2 (which the statement in our previous manuscript referred to), the Bayes Factor analysis indeed showed evidence ($BF_{01} > 3$) in favour of the null-hypothesis that the between-object encoding was no less strong than the within-objects encoding (Fig. 3f). We thank the reviewer for this very helpful comment.

Minor

Currently, report 2 only occurs in 50% of trials, whereas report 1 occurs in 100% of trials. In this way, unattended is conflated with less relevant, because the chance of being tested for the unattended item is only 50%. This may explain why people are worse – not because the item had been unattended, but because it was deemed less relevant for the task and therefore possibly strategically dropped / devoted less resources to.

The reviewer is correct, however, this is common practice in dual-retro cue tasks designed to induce “unattended” (or “deprioritized”) WM storage. For reference, please see seminal related neuroimaging studies in the field, e.g., Rose et al, 2016, *Science*; Christophel et al., 2018, *Nature Neuroscience*; Lewis-Peacock et al., 2012 *JoCN* (all cited in our manuscript). For more clarity, we added in the Introduction that the “unattended” manipulation may also be understood as “deprioritized”.

When comparing the repulsion vs. attraction RSA model-correlations with gaze, the author find that the repulsion model works better. However, the best model always seems to be the unbiased model, which is the in-between model. Leaving the unbiased model out of the comparison appears odd, and I am not sure how to interpret the repulsion vs. attraction comparison, if the middle-ground-model always works best.

Thank you for this comment, which we assume partly due to a display error of formulas in the previous PDF file for review – we sincerely hope the display error is now resolved (otherwise, the formulas are also shown in our preprint to the paper: <https://www.biorxiv.org/content/10.1101/2022.11.17.516917v1>).

A short answer is that the circle model is not the middle ground of repulsion and attraction. Averaging the repulsion and attraction (square-) geometries does not yield the unbiased circle geometry. Rather, attraction and repulsion distort the circle model in different ways (see equations on p. 21, which we hope are now displayed correctly in the PDF). Critically, the extreme repulsion and attraction (square-) RSA models are each equally dissimilar, in absolute terms, from the unbiased circle model, allowing us to compare them directly to infer whether gaze geometries are more attractively or repulsively biased. In contrast, comparing the square model(s) with the unbiased circle model has only limited meaning. The unbiased circle is always expected to be the best model among the three, unless participants’ gaze patterns would be **extremely** biased (e.g., B 's $\gg 0.5$) so that they would in fact resemble a “square” more than a circle. But this would seem unrealistic, also considering that the B values estimated from the behavioural response data were only around 0.1 to 0.2 (see Fig. 4b, behavioural modelling). We hope our explanations in the paper make these points clearer (together with the formulas), as we write on p. 12:

“If the gaze patterns were unbiased, we would expect both these ‘square’ models to correlate less well with the data than the unbiased (‘circle’) model with $B = 0$ (Fig. 4a *middle*). However, to the extent that the gaze patterns were repulsively biased, we would expect the repulsion model to outperform the attraction model, nearing (or in the case of extreme bias, even exceeding) the circle model (dashed

black in Fig. 4c). Contrasting repulsion and attraction models thus allowed us to quantify the extent of repulsive or attractive bias in the gaze patterns during the delay periods.”

Also, the authors never included cardinal orientations in their stimulus set. Might this contribute to the tendency for participants to repulse from cardinal orientations, strategically?

This is a thoughtful point, about which we can only speculate. Based on previous work, we have no reason to assume that cardinal repulsion would not also be observed if cardinal orientations were included in the stimulus set. For instance Bae (2019, Neuroimage) observed clear repulsion effects even if cardinal orientations were included. More generally, repulsive biases are well-documented in the behavioural WM literature and have been observed across a wide range of task conditions and materials (e.g., Taylor and Bias, 2018, J Neurosci).

The authors discuss their results in the context of related literature on microsaccades. While I agree this is sensible, the authors did minimal processing of their eye data, and did not quantify saccades. Fixational gaze behaviour is not only driven by microsaccades, but also by drifts and tremor. This may be acknowledged and/or could be quantified.

We are grateful for this comment. In the revision, we added a **direct quantitative analysis of microsaccadic activity** in the data (see newly added **Figure 5**), which lends support to our previous interpretation of the findings in terms of microsaccades. We feel that this extension strongly adds to the overall value and completeness of the study, and we hope the reviewer agrees. Thank you very much for this suggestion.

Fig 3: it might be nice to plot the object-specific and object-independent effects as their own time courses, to see the transition in time. Currently the reader has to imagine how the difference/average looks over time.

We agree. In the revision, we chose to combine this suggestion with the reviewer’s earlier point on evidence for no difference (between vs. within objects). Please see the newly added panels **Fig. 3e-f** which show the time courses of Bayes Factors (BF) for the critical difference (within-between). While the BF does not directly track changes in the absolute size of the difference (which can still be inferred from panels 3c-d), we think it nicely illustrates the

essential temporal dynamics of generalisation (note e.g., the crossing of time courses when the encoding of Stimulus 2 sets on). Thank you very much for encouraging us to include such an extension.

The first sentence of the abstract appears suboptimal. This currently reads as if this article is about neural decoding confounds. Rather, it is about representational formats in working memory, and how to track them through gaze.

Here, we only partially agree. Our study does build on a long history of M/EEG and fMRI studies decoding visual WM information from neural signal patterns. Our novel findings about (gaze-)representational formats aside, eye-movements are a number one source of signal distortions, at the very least when recording M/EEG, but to some extent also with other techniques (e.g., due to retinal displacement of visual input). We strongly anticipate that our empirical results will be received intensely (regardless of how we frame it in writing) also by the neuroimaging community, since the potential implications are self-evident, especially as the field is increasingly moving from 'only decoding' WM content to investigating its representational formats (e.g., neural representational geometries). We do agree, however, that our abstract should not suggest that we recorded neural data. We therefore edited the opening sentence to start with the eye-movements. Thank you for this hint.

There are some formatting issues, such as an empty formula and occasional symbols that seem out of place.

We apologise for this oversight. We have no explanation for the rendering problem in the previous PDF for review (the formulas have been displayed correctly before in our preprint to the paper, see link above). We do hope the problem is resolved now in the revised PDF.

Were eye-movements the primary outcome measure for this study, or was this study designed as an fMRI study, or alike? This may be good to disclose.

Eye-movements were the primary outcome measure (besides behavioural performance; see also Reporting Form) and the present study was designed specifically for this purpose. The idea emerged from preliminary observations during piloting other experiments in the lab, as well as from the published literature (see References in the manuscript).

In the introduction it is claimed how 1-item studies are typical in working memory research. I am not sure this is a fair depiction. For instance, what about all the studies on working memory capacity?

The reviewer is correct, this depiction was not ideal. We actually meant to refer more specifically to previous WM 'decoding' work (including MVPA of human fMRI or M/EEG, but also animal studies using single-cell and LFP recordings), where single-item testing has been very common. We revised the sentence and references accordingly.

In the first paragraph on the eye tracking results, the authors could make more explicit that this specific aspect (gaze biased by memory orientation) is consistent with several recent other studies that showed similar findings (the authors are aware of these studies).

Thank you, we added references to previous studies showing gaze biases with gratings oriented in 180° space (which we also cited in the Introduction) now also in this paragraph of the Results section.

Reviewer #2:

Remarks to the Author:

Review on Linde-Domingo & Spitzer: Geometry of visual working memory information in human gaze patterns

Overall evaluation: I am impressed by reading this manuscript reporting remarkable findings on the information that can be obtained from miniature eye movements during active fixation. The experimental procedures and the analyses seem sound, the writing is clear and succinct, and the conclusions are reasonable. I have only few questions and remarks. Congrats to the authors.

We are glad about the reviewer's very positive appraisal of our work.

Comments:

1) *It is difficult to understand how the authors could reconstructed the full angle of 360 degrees*

(2π), since there obviously an ambiguity in the mapping from the rotated to the original object. Therefore, I would have expected an orientation plot with re-rotation angles over a range of 180 degrees (π). Could you please clarify the explanations for Fig. 1 b & c?

We assume this to be just a misunderstanding, potentially because our example trial figures can only show a small selection of the (many) different orientations we used. We used pictures of real-world objects, which could indeed be rotated around the full circle (360° ; see Fig.1b and d for an illustration of all used orientations) without the ambiguities mentioned by the reviewer. For instance, 0° and 180° are in fact opposite angles with our stimuli (unlike e.g., with rotated gratings, for which these two angles would be the same).

2) *I am not convinced how the effect is generated. For example, when presented real-world object with a complex geometry as in the current experiment, it is hard to predict where the “visual center of gravity” is (where is the mean fixation position). Thus, the effect could be explained by tiny shifts in such an average fixation position from the presented location. Data to investigate such a hypothesis are there: What was the average fixation position during “Stimulus 1” and “Stimulus 2” epochs as a function of rotation angle? Did the authors observe any biases here? If yes, then an explanation of the effect could be independent of the orientation, but merely be remembering the original fixation bias during the perceptual encoding.*

This is an interesting point, which we believe our revision sheds important new light on. We now included a **direct analysis of microsaccadic activity**, which was also encouraged by the reviews. The additional results (see newly added Figure 5) support our previous speculations about (potentially unintentional) microsaccades, i.e. short, jerk-like eye-movements near fixation. This additional result renders it less likely that participants merely remembered an original (static) fixation bias. Furthermore, the average gaze bias during the stimuli (to be precise, they mostly occurred only shortly *after* the stimuli, see Fig. 1c) were not fully identical to those observed later during WM – the gaze patterns became increasingly more object-independent (Fig. 3) and repulsively biased (Fig. 4), either of which would be harder to explain if participants had merely remembered an original fixation bias.

3) *In the discussion, the authors relate the current findings to previous work on microsaccades and attentional effect. However, there is no reason to speculate on the microsaccadic effects, given that the data are recorded and microsaccadic activity could be analyzed to clarify the picture. This would be a useful extension of the current analyses. This point is not critical for a potential revision, but could give a more complete view into the gaze dynamics during the different epochs of the*

experiment.

We are thankful for this comment, which encouraged us to include the analysis of microsaccadic activity in Fig. 5 as mentioned above (see our previous responses). The additional results fully support our previous interpretation and, in our view, they strongly add to the overall value of the study. Thank you very much!

4) *In my PDF, some of the equations and formula characters were not readable (mainly due to missing characters). Therefore, I could not check their correctness.*

We apologise and sincerely hope the formulas in the revised manuscript are now rendered correctly. Should the problem still persist, we also provide the link to our preprint (<https://www.biorxiv.org/content/10.1101/2022.11.17.516917v1>), where all formulas are displayed intact.

Reviewer #3:

Remarks to the Author:

In the manuscript NATHUMBEHAV-23010084 entitled „Geometry of visual working memory information in human gaze patterns” the authors describe the results of one experiment in which they analyze miniature eye movements (defined as saccadic eye movements smaller than 1° of visual angle) during a dual retro-cue task testing orientations of rotated objects. Their results show that miniature eye movements contain rich information on the representational format in visual working memory. During stimulus presentation, eye movement patterns were in line with orientation encoding and changed to more object-independent formats during cued maintenance. The more attention had been withdrawn from one stimulus, the more abstract the representational format. Even more, during periods of unattended storage, categorical reporting biases emerged.

Using eye movements to study visual working memory is an advancing field. The manuscript provides an excellent example of the usefulness of studying eye movements to investigate memory processes (i.e. the nature and time course of representation in visual working memory) and seems suitable for publication in Nature Human Behavior when appropriately addressing my concerns listed below.

We thank the reviewer for these very kind words.

Overall, the presented experiment is carefully conducted, and the analyses are of highest quality. I wonder a bit why they aggregated the data prior to the frequentist analyses and did not use hierarchical analyses that would allow to account for interindividual variability in the strength of the observed effects. I had some problems understanding the calculation of rho and pcluster. I would wish the authors to describe the calculation of these analyses more carefully. Maybe, I missed an important piece as not all formulas were displayed correctly in the available pdf (p. 19, l. 642, 651, 653 onwards).

We apologise for the display error in the previous PDF file for review. We hope things become clearer now that the formulas are (hopefully) all rendered correctly. Otherwise please see our preprint (<https://www.biorxiv.org/content/10.1101/2022.11.17.516917v1>) for the complete formulas.

We used a rather basic approach in the behavioural analysis, since the modelling was mostly intended to quantify ('measure') the strength of the repulsive bias, which was clearly visible even directly in the descriptive data (Fig. 4b; see also the descriptively excellent fit of the model to these data). We neither aimed at identifying a best model for this well-known effect, nor at making inference about potentially underlying neural computations – we only used the modelling to compare the strength of bias (or lack thereof) across conditions. The same goal could also be achieved with even simpler, merely descriptive measures of bias, but we decided for the present approach because it at the same time integrates seamlessly with our novel, geometry-based approach to eye-tracking analysis (Fig. 4).

Rho refers to the (Pearson) correlation coefficient that was computed in RSA (see Fig. 2b) – we edited the descriptions throughout the manuscript to make this clearer. We apologise that this was missing in the previous manuscript. For cluster-based statistics (p-cluster) we used a well-established approach described by Maris and Oostenveld (2007) and we added a short description of the approach in the revised Methods section (on p. 23, end of the 1st paragraph). We thank the reviewer for encouraging these improvements for clarity.

Eye movements are guided by various top-down and bottom-up factors. Consequently, eye movements and especially miniature eye movements are sometimes very noisy measures of human behavior (even in such a controlled experimental environment as in the present experiment). In addition, the authors aim to establish miniature eye movements and their analyses approach as a tool to get insights into visual working memory. To be more convincing that the results would replicate and generalize to other task typically used to study visual working memory, I would suggest the authors show the same set of

effects in a new experiment. For instance, a typical visual working memory task is to remember the color of squares presented at different screen locations. If colors were arranged in a circle, the same analysis strategy may be applicable.

This is a thoughtful suggestion. One may indeed expect systematic gaze biases to stimulus *locations* in a task as sketched by the reviewer. However, in our view, this expectation follows not primarily from our current study of WM for object *orientations*, but follows much more directly from accumulating previous work, which has already shown gaze bias to the *location* of WM stimuli to occur (De Vries & Van Ede, 2023; Liu et al., 2022; van Ede et al., 2019). The reviewer is correct that it would be technically feasible to apply our RSA approach also to circularly arranged locations, and location encoding in gaze might also register as such in our RSA model. Critically, however, one would not be able to examine object-specific vs object-independent (resp. “concrete” vs “abstract”) patterns in such a hypothetical experiment, unlike with our oriented objects paradigm which was specially designed for our present study goals (see Introduction). Since basic gaze biases to the *location* of WM stimuli have already been established (and replicated) in recent literature, we hope the reviewer approves that we did not add another location-based experiment to our present study.

The figures are very well designed, however in some cases, they are too small. For instance, the polar plot in Figure 4b is very small. Furthermore, I miss the description of the error bands in the plots with timeline data (e.g., Figure 4c, Figure 3c). I cannot see purple markers below the plots in Figure 4c. Some plots have a red line at the bottom.

Thank you very much, we improved the figures accordingly (see revised Fig. 4b, now with the polar plot enlarged) and carefully checked/edited all figure legends for completeness and correctness. The error bands reflect SEM. We made sure that this is now explicitly stated in each of the figure captions.

I do not quite agree with the criticism that the results are at odds with the idea of a single-item focus of attention. Only one item can be in the focus of attention, however several items can be in an activated state in working memory and held active with maintenance strategies. The data is highly aggregated but on the trial level one can only look at one position at a time. Thus, the authors should be more specific at how their results are at odds with these current theories of working memory.

Please see our edits of this paragraph in the revised Discussion (p. 16). We in fact did not mean to suggest that our findings would contradict a 1-item focus, but we were anticipating that some readers might interpret them as such (specifically, the result from single-trial analysis in Fig. 2e), which we sought to put into perspective by offering alternative explanations. We edited the section to avoid misunderstandings:

“At face value, the temporary simultaneity of both WM contents (Fig. 2e), in a putative index of attention, might appear to be at odds with the idea of an exclusive single-item focus of attention in WM (Oberauer, 2002; Olivers et al., 2011; but see Beck et al., 2012; Zhang et al., 2018). However, another possible interpretation is that the (re)allocation of attention to different stimuli (or tasks) in WM may take time to complete. For example, (...)”

For completeness, the reviewer is correct that one can look only at one position at a time, however, this position can nevertheless be a function of more than one variable (e.g., in our case of the two stimuli's orientations). Our results in Fig. 2e suggest that the gaze position associated with a given Stimulus 2 orientation varied additionally with the orientation of Stimulus 1. While the noise level in the data prevents us from illustrating the results for all possible combinations of orientations (16 x 16) individually, the full pattern suggested by our analysis could be imagined as a “ring” (driven by the orientation of Stimulus 2) “of circles” (each driven by the orientation of Stimulus 1).

Interestingly, our newly added microsaccade analysis may raise yet another interesting possibility to be investigated in future work: that microsaccades might go back and forth between concurrently maintained WM contents *within trials*. This possibility might be investigated in future studies with tailored designs (e.g., using higher WM loads and longer periods of concurrent maintenance than in our present work).

We again thank the reviewers for their excellent comments and suggestions, and we await further directions, including from the editor, on how to proceed with our paper.

Decision Letter, first revision:

18th August 2023

Dear Dr. Linde-Domingo,

Thank you for your patience as we've prepared the guidelines for final submission of your Nature Human Behaviour manuscript, "Geometry of visual working memory information in human gaze patterns" (NATHUMBEHAV-23010084A). Please carefully follow the step-by-step instructions provided in the attached file, and add a response in each row of the table to indicate the changes that you have made. Please also address the additional marked-up edits we have proposed within the reporting summary. Ensuring that each point is addressed will help to ensure that your revised manuscript can be swiftly handed over to our production team.

We would hope to receive your revised paper, with all of the requested files and forms within two-three weeks. Please get in contact with us if you anticipate delays.

If you have not done so already, please alert us to any related manuscripts from your group that are under consideration or in press at other journals, or are being written up for submission to other journals (see:

<https://www.nature.com/nature-research/editorial-policies/plagiarism#policy-on-duplicate-publication> for details).

Nature Human Behaviour offers a Transparent Peer Review option for new original research manuscripts submitted after December 1st, 2019. As part of this initiative, we encourage our authors to support increased transparency into the peer review process by agreeing to have the reviewer comments, author rebuttal letters, and editorial decision letters published as a Supplementary item. When you submit your final files please clearly state in your cover letter whether or not you would like to participate in this initiative. Please note that failure to state your preference will result in delays in accepting your manuscript for publication.

In recognition of the time and expertise our reviewers provide to Nature Human Behaviour's editorial process, we would like to formally acknowledge their contribution to the external peer review of your manuscript entitled "Geometry of visual working memory information in human gaze patterns". For those reviewers who give their assent, we will be publishing their names alongside the published article.

Cover suggestions

As you prepare your final files we encourage you to consider whether you have any images or illustrations that may be appropriate for use on the cover of Nature Human Behaviour.

ORCID

Non-corresponding authors do not have to link their ORCIDs but are encouraged to do so. Please note that it will not be possible to add/modify ORCIDs at proof. Thus, please let your co-authors know that if they wish to have their ORCID added to the paper they must follow the procedure described in the following link prior to acceptance:

Nature Human Behaviour has now transitioned to a unified Rights Collection system which will allow our Author Services team to quickly and easily collect the rights and permissions required to publish your work. Approximately 10 days after your paper is formally accepted, you will receive an email in providing you with a link to complete the grant of rights. If your paper is eligible for Open Access, our Author Services team will also be in touch regarding any additional information that may be required to arrange payment for your article.

Please note that *Nature Human Behaviour* is a Transformative Journal (TJ). Authors may publish their research with us through the traditional subscription access route or make their paper immediately open access through payment of an article-processing charge (APC). Authors will not be required to make a final decision about access to their article until it has been accepted. Find out more about Transformative Journals

Authors may need to take specific actions to achieve compliance with funder and institutional open access mandates. If your research is supported by a funder that requires immediate open access (e.g. according to Plan S principles) then you should select the gold OA route, and we will direct you to the

compliant route where possible. For authors selecting the subscription publication route, the journal's standard licensing terms will need to be accepted, including self-archiving policies. Those licensing terms will supersede any other terms that the author or any third party may assert apply to any version of the manuscript.

[REDACTED]

Best regards,
Alex McKay
Editorial Assistant
Nature Human Behaviour

On behalf of

Giacomo Ariani
Editor
Nature Human Behaviour

Reviewer #1:

Remarks to the Author:

I appreciate the elaborate responses to my previous comments. I remain curious to the future outcomes of further experiments, but also appreciate how this could be conceived to lie beyond the scope of the current paper that already goes beyond the state of the art in terms of findings and analysis arsenal. I have no further comments.

Reviewer #2:

Remarks to the Author:

I am super happy with the revised version of the manuscript; the additional analyses clarify my most important issues. Congrats to the exciting work!

Reviewer #3:

Remarks to the Author:

The authors sufficiently addressed my concerns. I only have one minor comment. I think, "sans" (Introduction, first paragraph) may be replaced by "without".

Author Rebuttal, first revision:

We want to thank all reviewers for their dedicated effort in reviewing our manuscript.

Reviewer #1:

Remarks to the Author:

I appreciate the elaborate responses to my previous comments. I remain curious to the future outcomes of further experiments, but also appreciate how this could be conceived to lie beyond the scope of the current paper that already goes beyond the state of the art in terms of findings and analysis arsenal. I have no further comments.

We thank the reviewer very much for the positive response, and we hope our present study will inspire such future work.

Reviewer #2:

Remarks to the Author:

I am super happy with the revised version of the manuscript; the additional analyses clarify my most important issues. Congrats to the exciting work!

Thank you very much, we are very glad about this response.

Reviewer #3:

Remarks to the Author:

The authors sufficiently addressed my concerns. I only have one minor comment. I think, "sans" (Introduction, first paragraph) may be replaced by "without".

We are pleased by the reviewer's positive responses. We also agree with their last remaining suggestion and changed the term to "without". Thank you very much.

Final Decision Letter:

Dear Dr Linde-Domingo,

We are pleased to inform you that your Article "Geometry of visuospatial working memory information in miniature gaze patterns", has now been accepted for publication in *Nature Human Behaviour*.

Please note that *Nature Human Behaviour* is a Transformative Journal (TJ). Authors may publish their research with us through the traditional subscription access route or make their paper immediately open access through payment of an article-processing charge (APC). Authors will not be required to make a final decision about access to their article until it has been accepted. Find out more about Transformative Journals

With best regards,

Giacomo Ariani
Editor
Nature Human Behaviour